# On Image Segmentation With Noisy Labels: Characterization and Volume Properties of the Optimal Solutions to Accuracy and Dice

**Marcus Nordström** *
Department of Mathematics
KTH Royal Institute of Technology
Stockholm, Sweden
marcno@kth.se

**Henrik Hult**
Department of Mathematics
KTH Royal Institute of Technology
Stockholm, Sweden
hult@kth.se

**Jonas Söderberg**
Department of Machine Learning
RaySearch Laboratories
Stockholm, Sweden
jonas.soderberg@raysearchlabs.com

**Fredrik Löfman**
Department of Machine Learning
RaySearch Laboratories
Stockholm, Sweden
fredrik.lofman@raysearchlabs.com

## Abstract

We study two of the most popular performance metrics in medical image segmentation, Accuracy and Dice, when the target labels are noisy. For both metrics, several statements related to characterization and volume properties of the set of optimal segmentations are proved, and associated experiments are provided. Our main insights are: (i) the volume of the solutions to both metrics may deviate significantly from the expected volume of the target, (ii) the volume of a solution to Accuracy is always less than or equal to the volume of a solution to Dice and (iii) the optimal solutions to both of these metrics coincide when the set of feasible segmentations is constrained to the set of segmentations with the volume equal to the expected volume of the target.

## 1 Introduction

One of the most central problems in medical image analysis is to identify the region of an image associated with a certain target structure. This problem, referred to as image segmentation or delineation, is often very time consuming to solve manually. Consequently, there is great interest in the development of methods that can assist in automation of the procedure. Since 2015, it has become increasingly popular to address the segmentation problem using machine learning based methods, and in particular, fully convolutions neural networks with U-net architecture. Such methods commonly dominate the winning submissions to segmentation contests and are backed by a large base of supporting literature [6, 8, 13, 19, 28, 32].

Despite the success, performance of these methods will, like any machine learning method, depend on the quality of the available data [4, 37]. Since it is well known that the data commonly used in practice is produced by medical practitioners that delineate structures in an inconsistent manner [25], it is important to understand the impact of label noise. One way to study the influence of label noise is to consider how the noise impacts segmentations that are theoretically optimal with respect to the metric used for measuring performance. Even if theoretically optimal solutions may not be

---

*Author is also affiliated with RaySearch Laboratories.

36th Conference on Neural Information Processing Systems (NeurIPS 2022).

attainable in practice, studying them gives important insights into what the effect label noise has on the particular metric and any training method that is designed to maximize it.

Arguably, the most simple and classical choice of metric is Accuracy; the fraction of the image that is correctly delineated. This metric is most commonly targeted by taking a $1/2$-threshold of predictions from a model trained with the cross-entropy loss [24]. However, since Accuracy may not reflect the desired behaviour when the data is unbalanced, that is, when the target structure is much smaller than the background, alternative metrics are often preferred. The most popular such alternative is the Sørensen-Dice coefficient, or Dice for short and is related to the $F_1$-metric used in binary classification. This metric is most commonly targeted by taking a $1/2$-threshold of predictions from a model trained with the soft-Dice loss, a smoothed version of the Dice metric [24]. Other examples considered in the literature include the Jaccard index and variations of the Haussdorff metric [33].

In this work, we conduct a theoretical investigation of the effect label noise has on optimal segmentations with respect to the performance metrics Accuracy and Dice. Because the volume of a proposed segmentation may be used for important properties such as estimating the size of a tumor [9], we pay special attention to the effect noise may have on the volume of the optimal segmentations.

**Contributions:** A characterization of all optimal solutions to Accuracy and Dice when the target is noisy is provided. This characterization is used to analyze the volume of the optimal solutions and we prove: (i) sharp upper and lower bounds on the volume of optimal segmentations with respect to Accuracy and Dice, (ii) that the volume of the optimal solutions to Accuracy always is less or equal to the volume of the optimal solutions to Dice and (iii) that the optimal solutions to both metrics coincide when the volume is held fix. We also show the relevance of the problem in a practical setting by including experiments on data from the Gold Atlas project [25] and The Lung Image Database Consortium (LIDC) and Image Database Resource Initiative (IDRI) [1].

## 2    Related work

Deep learning methods are playing an increasingly vital part in the development of medical image analysis. However, deep learning models require large annotated data sets for successful training that rarely are available in the clinics. Even if some data sets are being curated for training deep learning models, it is generally difficult and expensive to accurately annotate large collections of medical images. Moreover, training data may include corrupted or noisy labels. This is particularly the case in image segmentation where different annotators may have different views on the correct delineation of a region of interest, leading to uncertainty about the true label, see e.g. [3, 23]. Noisy labels may also appear due to automated systems or non-expert systems being used to annotate large volumes of data, see [5, 27].

There is a large body of literature on the impact of label noise in image segmentation, see [34] and [14] for recent reviews. The proposed solutions to limit the loss of performance when the labels are noisy include label cleaning and pre-processing, e.g. [10], modification of network architechtures, e.g. [36], robustification of loss functions, e.g. [21], reweighting of training data, see [22, 39], and many others. These approaches are of practical nature and generally address methodology that improve the performance on some chosen noisy data set. On the contrary, the literature that address the effect of label noise from a theoretical point of view in the context of image segmentation is rather limited. It was shown that the loss function soft-Dice, in contrast to the loss function cross-entropy, does not yield optimal predictions that coincide with the pixel-wise marginals and that the associated volume is biased [2]. This motivated methods for post-calibrating uncalibrated marginal estimates [29] and a more general investigation of the relationship between volume and marginal calibration [26]. Finally, an alternative volume preserving segmentation method based on optimal transport theory was proposed and investigated in [20].

Another domain of related work can be found in the binary classification literature. The connection between the study of solutions to Accuracy and Dice in segmentation and the study of Accuracy and $F_1$ in binary classification was investigated in [24]. Of importance to us are *optimal plug-in* classifiers to Accuracy and $F_1$. That is, classifiers that are obtained by processing the posterior class probabilities or estimates thereof using a threshold. For Accuracy, this relates to the classical Bayes classifier which has been studied since the origin of the field [35]. Early work on the existence of

such a classifier for $F_1$, and the fact that this threshold is lower than the threshold for Accuracy can be traced to [38]. Further work showed this threshold to be equal to half of the maximal attainable $F_1$-score [18]. Lots of extensions of these works have been proposed, but to the best of our knowledge no such extension is in the direction of our work.

## 3   Preliminaries

In our work we find that it is convenient to do the theoretical analysis over a continuous domain and where all encountered continuous spaces are equipped with their associated Borel $\sigma$-fields. Formally, let $\Omega = [0,1]^n \subset \mathbb{R}^n$ be the unit cube of dimension $n \geq 1$ and $\lambda$ be the associated standard normalized Lebesgue measure such that $\lambda(\Omega) = 1$. The space of segmentations is denoted by $\mathcal{S}$ and is formally given by the space of measurable functions from $\Omega$ to the binary numbers $\{0, 1\}$. For any segmentation $s \in \mathcal{S}$, $s(w) = 1$ implies that the object of interest occupies the site $\omega \in \Omega$. In a numerical setting, a discretization of the domain $\Omega$ is commonly used. For instance, 3D CT scans are often represented by a three dimensional voxel grid of an approximate order of $512 \times 512 \times 100$. In such situations the space of segmentations are given by all possible binary functions defined on this voxel grid. Note that any segmentation on a discretized domain can be incorporated in our continuous framework by using appropriate step functions. Details on this can be found in the end of Section 4.

Beyond the space of segmentations, several other technical constructions are introduced. This includes the space of measurable functions from $\Omega$ to $[0,1]$ which we denote by $\mathcal{M}$ and refer to as the *marginal functions*. We also let $\|f\|_1 \doteq \int_\Omega |f(\omega)| \lambda(d\omega)$ and $\bar{f} \doteq 1 - f$, where $f$ is any measurable function defined on $\Omega$. Throughout we adopt the convention that two $\lambda$-measurable functions $f, g$ are equal if they are equal $\lambda$-a.e. We will use $I\{\cdot\}$ to denote the identity function, and when $F$ is a cumulative distribution function, we will denote the left limit $F(t-) = \lim_{s \uparrow t} F(s)$. Finally, for a given volume $v \in [0,1]$, we let $\mathcal{S}_v \doteq \{s \in \mathcal{S}, \|s\|_1 = v\}$ be the set of segmentations with volume $v$.

Classically, metrics in medical image segmentation are defined per image as functionals over two deterministic segmentations [33]. When noise is present, the label becomes a random variable and the metrics need to be extended to a functional over one deterministic segmentation and one random label segmentation. In this work, the soft labeling convention for this extension is adopted [11, 15, 16, 17, 31]. Also, since we do our analysis with respect to measurable functions on a continuous domain instead of functions on a finite index set, we replace the sums usually used with integrals. This, however, does not change the intuition of the metrics, and the definition using sums can be seen a special case.

**Definition 1.** *For any $m \in \mathcal{M}$, Accuracy is given by*

$$\mathrm{A}_m(s) \doteq \int_\Omega [s(\omega)m(\omega) + \bar{s}(\omega)\bar{m}(\omega)]\lambda(d\omega), \quad s \in \mathcal{S} \tag{1}$$

**Definition 2.** *For any $m \in \mathcal{M}$, Dice is given by*

$$\mathrm{D}_m(s) \doteq \frac{2\int_\Omega s(\omega)m(\omega)\lambda(d\omega)}{\|s\|_1 + \|m\|_1}, \quad s \in \mathcal{S}. \tag{2}$$

For a noisy segmentation $L$, that is, a random variable taking values in $\mathcal{S}$, $m$ can be taken to be the exact marginal success probability $m(\omega) = \mathbb{E}[L(\omega)]$, $\omega \in \Omega$. Such marginal functions are important in theory but can rarely be obtained in practice. Alternative choices of marginal functions include finite sample approximations, that is, point-wise averages over finite observations of $L$, and estimates of $\mathbb{E}[L]$ according to a single annotator [11, 15, 16, 17, 31]. These choices of $m$ are important because they are sometimes used for training machine learning models. Finally, note that $\mathrm{A}_{\mathbb{E}[L]}(s) = \mathbb{E}[\mathrm{A}_L(s)]$ and $\mathbb{E}[\mathrm{A}_L(s)]$ is a common alternative way of specifying the metric. For Dice, this sort of relationship does not hold in general $\mathrm{D}_{\mathbb{E}[L]}(s) \neq \mathbb{E}[\mathrm{D}_L(s)]$. However, it does hold that $\mathrm{D}_{\mathbb{E}[L]}(s) = \mathbb{E}[\mathrm{D}_L(s)]$ when the volume of the noisy labels is constant $\mathrm{Var}[\|L\|_1] = 0$, and it holds approximately $\mathrm{D}_{\mathbb{E}[L]}(s) \approx \mathbb{E}[\mathrm{D}_L(s)]$ when the variance of the volume of the noisy labels is small $\mathrm{Var}[\|L\|_1] \approx 0$, which is often the case in medical image segmentation applications. Examples of observations of a particular $L$ for a couple of different target structures are depicted in Figure 1.

Because of the fact that $\|m\|_1 = \mathbb{E}[\|L\|_1]$ when $m$ is taken to be the exact marginal success probability, $\|m\|_1$ plays an important role in the medical segmentation context, either theoretically as

the expected volume of the target or as an approximation thereof. Understanding how $\|m\|_1$ relates to $\|s\|_1$, where $s$ is an optimal segmentation to Accuracy or Dice will be central in our work. To the best of our knowledge, this has not been studied in prior work.

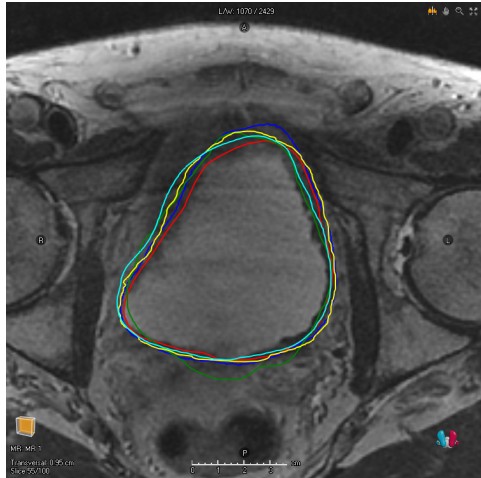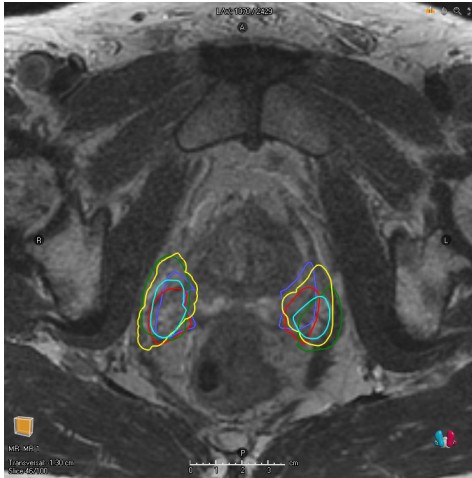

Figure 1: To the left is Urinary bladder and to the right is Neurovascular bundles for one patient in the Gold Atlas data [25]. Each line is associated with the boundary of a segmentation produced by a particular annotator. The screenshots are taken with RayStation 12A (RaySearch Laboratories AB, Stockholm Sweden).

## 4    Main results

The objective of our analysis is to characterize the optimizers to $A_m$ and $D_m$ and give a detailed description of the volume of the optimal segmentations. That is, for a given $m \in \mathcal{M}$, identify properties (e.g. volume) of the optimal segmentation $s \in \mathcal{S}$ that maximimize Accuracy or Dice. To this end, consider the probability measure on $[0, 1]$ given by the push-forward measure $\lambda \circ \bar{m}^{-1}(\cdot)$ and let $F_m$ denote its cumulative distribution function,

$$F_m(t) \doteq \lambda \circ \bar{m}^{-1}([0, t]) = \int_\Omega I\{\bar{m}(\omega) \leq t\}\lambda(d\omega), \quad t \in [0, 1]. \tag{3}$$

The function $F_m(t)$ may be interpreted as the volume of the set of sites with non-success probability less than or equal to $t$. In other words, the volume of the sub-level set of $\bar{m}$ at level $t$.

Since $F_m$ is the cumulative distribution function of a probability distribution on $[0, 1]$, it has several well-known properties making it easy to work with, e.g., $F_m$ is non-decreasing and right-continuous with $F_m(1) = 1$. Of particular interest to us is that it has a generalized inverse given by

$$F_m^{-1}(v) \doteq \inf\{t : F_m(t) \geq v\}, \quad v \in [0, 1], \tag{4}$$

which can be interpreted as the minimum level at which the volume of the corresponding sub-level set of $\bar{m}$, is at least $v$. This function is often referred to as the quantile function and also has several well known properties; it is non-decreasing and left-continuous. Moreover, it allows us to define the following important class of segmentations for a given $m \in \mathcal{M}$.

$$\mathcal{S}_{m,v} \doteq \left\{ s \in \mathcal{S}_v \;\middle|\; \begin{array}{l} \int_\Omega s(\omega)I\{m(\omega) < 1 - F_m^{-1}(v)\}\lambda(d\omega) = 0, \\ \int_\Omega \bar{s}(\omega)I\{m(\omega) > 1 - F_m^{-1}(v)\}\lambda(d\omega) = 0, \end{array} \right\}, \quad v \in [0, 1]. \tag{5}$$

The described class $\mathcal{S}_{m,v}$ is informally the set of segmentations with volume $v$ that assigns 1 to sites $\omega$ where $m(\omega)$ is large. If $t = F_m^{-1}(v)$ is a continuity point of $F_m$, i.e. $\lambda(\omega : \bar{m}(\omega) = t) = 0$, then $\mathcal{S}_{m,v}$ only consist of the elements that are $\lambda$-a.e. equal to the segmentation $s(\omega) = I\{m(\omega) \geq 1 - F_m^{-1}(v)\}$.

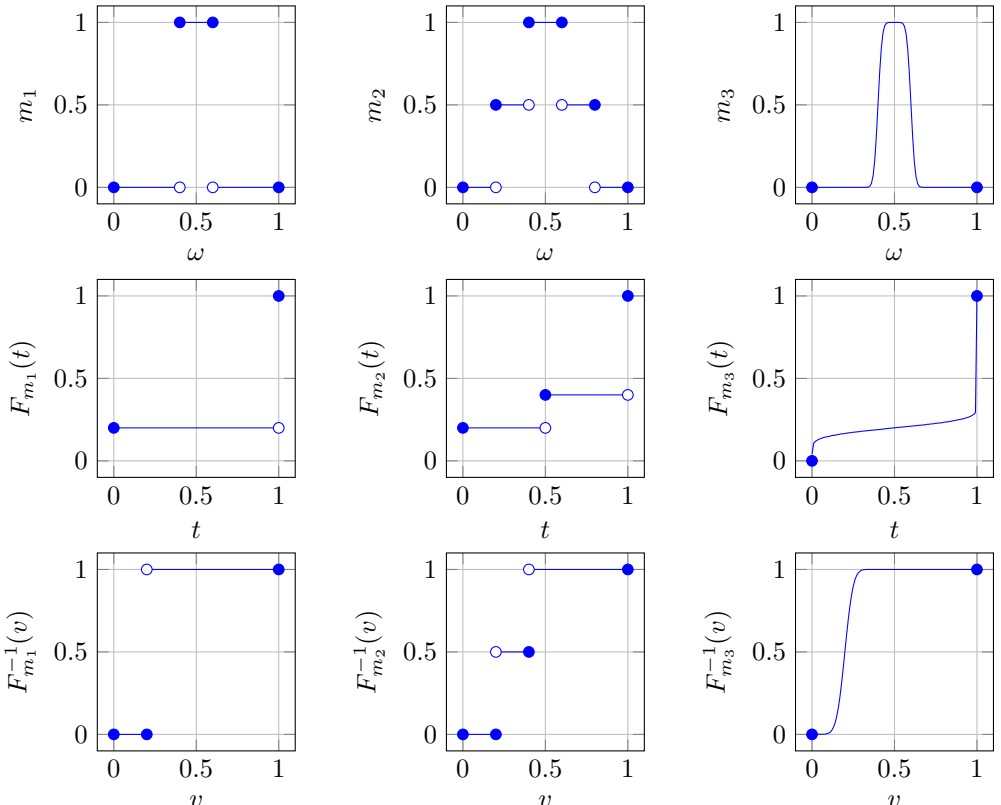

Figure 2: Three examples of marginal functions $m_1$, $m_2$, $m_3$ on a one-dimensional domain $\omega \in \Omega = [0,1]$ together with the associated $F_{m_1}, F_{m_2}, F_{m_3}$ and $F_{m_1}^{-1}, F_{m_2}^{-1}, F_{m_3}^{-1}$. The case $m_1$ occurs when there is no noise. The case $m_2$ can for instance occur when the segmentation is given by $I\{0.2 \leq \omega \leq 0.6\}$ with probability $0.5$ and $I\{0.4 \leq \omega \leq 0.8\}$ with probability $0.5$. The case $m_3$ can occur when the edges of the segmentation are randomly perturbed by some continuous random variable.

A lot of our analysis can be simplified if $F_m$ is assumed to be invertible almost everywhere. However, this would require $m$ to not have any non-neglible constant regions, which for instance excludes any $m$ that is given by an empirical approximations using a finite number of samples. Consequently, in the sequel, we treat general $F_m$. Our first result contains the essential ingredients for characterizing the optimizers to Accuracy and Dice.

**Lemma 1.** *For any $m \in \mathcal{M}$ and $v \in [0,1]$*

$$\sup_{s \in \mathcal{S}_v} \int_\Omega s(\omega)m(\omega)\lambda(d\omega) = \int_0^v (1 - F_m^{-1}(u))du, \tag{6}$$

*and the elements where the supremum is attained is given by $\mathcal{S}_{m,v}$.*

A complete proof is given in the Supplementary Document and outlined as follows. The first part shows that the class $\mathcal{S}_{m,v}$ is the class of optimal solutions by showing that for any $s^* \in \mathcal{S}_{m,v}$ and $s \in \mathcal{S}_v \setminus \mathcal{S}_{m,v}$, $\int_\Omega (s^*(\omega) - s(\omega))m(\omega)\lambda(d\omega) > 0$. The second part proves the equality (6) using an application of the quantile transform. That is, if $U$ has uniform distribution on $[0,1]$ then $F_m^{-1}(U)$ has cdf given by $F_m$ and $\int_0^1 tF_m(dt) = \mathbb{E}[F_m^{-1}(U)] = \int_0^1 F_m^{-1}(u)du$.

Lemma 1 allows us to reduce the constrained optimization problem over the rather complicated space of segmentations, to a one-dimensional integral with respect to the quantile function. It is the starting point for our analysis.

In the remaining section our main theoretical results are presented. In Theorem 1, we provide a characterization of all of the optimal solutions to Accuracy based on volume. In addition, sharp upper and lower bounds on the volume of the associated optimal segmentations are provided.

**Theorem 1.** *For any $m \in \mathcal{M}$, the class of maximizers to $\mathrm{A}_m$ is given by $\cup_{v \in \mathcal{V}^{\mathrm{A}m}} \mathcal{S}_{m,v}$ where*

$$\mathcal{V}^{\mathrm{A}m} \doteq [F_m(1/2-), F_m(1/2)]. \tag{7}$$

*Moreover, $\mathcal{V}^{\mathrm{A}m}$ satisfies the following bounds*

$$\mathcal{V}^{\mathrm{A}m} \subseteq [\max\{2\|m\|_1 - 1, 0\}, \min\{2\|m\|_1, 1\}], \tag{8}$$

*where the bounds are sharp in the sense that there for any $v \in [0,1]$ exist $m_0, m_1 \in \mathcal{M}$ such that $\|m_0\|_1 = \|m_1\|_1 = v$ and*

$$\inf \mathcal{V}^{\mathrm{A}m_0} = \max\{2\|m_0\|_1 - 1, 0\}, \quad \sup \mathcal{V}^{\mathrm{A}m_1} = \min\{2\|m_1\|_1, 1\}. \tag{9}$$

The complete proof is given in the Supplementary Document and outlined as follows. First, the function

$$a_m(v) \doteq v + 1 - \|m\|_1 - 2 \int_0^v F_m^{-1}(u)du, \quad v \in [0,1], \tag{10}$$

is introduced and then Lemma 1 is used to show that $\sup_{s \in \mathcal{S}} \mathrm{A}_m(s) = \sup_{v \in [0,1]} a_m(v)$. Consequently, the class of optimal solutions to $\mathrm{A}_m$ is given by $\cup_{v \in \mathcal{V}^{\mathrm{A}m}} \mathcal{S}_{m,v}$, where $\mathcal{V}^{\mathrm{A}m}$ is the set of optimizers to $a_m$. The rest of the proof consists of detailed analysis of $a_m$ and is composed of three parts. The first part is to show (7) by finding one optimal solution and then identifying all volumes that yield the same optimal value. The second part is to provide the lower and upper bounds on the elements of $\mathcal{V}^{\mathrm{A}m}$ in terms of $\|m\|_1$ given by (8). The third part is to provide examples of situations where the extreme cases occur (9). In Figure 3, a case that is extreme both in the lower and in the upper sense is illustrated.

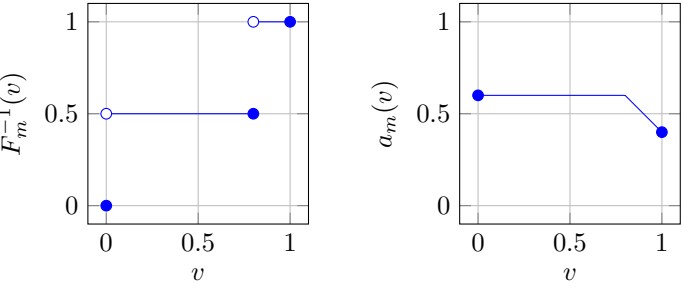

Figure 3: To the left is a particular quantile function $F_m^{-1}$ and to the right is the associated function $a_m$ given by (10). Here $F_m^{-1}(v) = \frac{1}{2}I_{(0,2\|m\|_1]}(v) + I_{(2\|m\|_1, 1]}(v), v \in [0,1]$ with $\|m\|_1 = 0.4$ that satisfies $\mathcal{V}^{\mathrm{A}m} = [0, 2\|m\|_1] = [\max\{2\|m\|_1 - 1, 0\}, \min\{2\|m\|_1, 1\}]$ and consequently is an extreme case to (9) in both the lower sense and the upper sense.

In Theorem 2, we provide a characterization of all of the optimal solutions to Dice based on volume. In addition, sharp upper and lower bounds on the volume of the associated optimal segmentations are provided.

**Theorem 2.** *For any $m \in \mathcal{M}$, the class of maximizers to $\mathrm{D}_m$ is given by $\cup_{v \in \mathcal{V}^{\mathrm{D}m}} \mathcal{S}_{m,v}$ where*

$$\mathcal{V}^{\mathrm{D}m} \doteq [F_m((1 - \sup_{s \in \mathcal{S}} \mathrm{D}_m(s)/2)-), F_m(1 - \sup_{s \in \mathcal{S}} \mathrm{D}_m(s)/2)]. \tag{11}$$

*Moreover, $\mathcal{V}^{\mathrm{D}m}$ satisfies the following bounds*

$$\mathcal{V}^{\mathrm{D}m} \subseteq [\|m\|_1^2, 1], \tag{12}$$

*where the bounds are sharp in the sense that there for any $v \in (0,1]$ exist $m_0, m_1 \in \mathcal{M}$ such that $\|m_0\|_1 = \|m_1\|_1 = v$ and*

$$\inf \mathcal{V}^{\mathrm{D}m_0} = \|m\|_1^2, \quad \sup \mathcal{V}^{\mathrm{D}m_1} = 1. \tag{13}$$

The complete proof is given in the Supplementary Document and outlined as follows. First, the function

$$d_m(v) \doteq \frac{2 \int_0^v (1 - F_m^{-1}(u))du}{\|m\|_1 + v}, \quad v \in [0,1], \tag{14}$$

is introduced and then Lemma 1 is used to show that $\sup_{s \in \mathcal{S}} D_m(s) = \sup_{v \in [0,1]} d_m(v)$. Consequently, the class of optimal solutions to $D_m$ is given by $\cup_{v \in \mathcal{V}^{D_m}} \mathcal{S}_{m,v}$, where $\mathcal{V}^{D_m}$ is the set of optimizers to $d_m$. The remaining proof consists of detailed analysis of $d_m$ and composed of three parts. The first part is to show (11) which is derived by careful investigation of the properties of the function $\delta(v) = \frac{(\|m\|_1 + v)^2}{2} \partial_v d_m(v)$, which has the same sign as $\partial_v d_m(v)$ and therefore can be used to identify optimal values of $d_m$. The second part is to provide the lower and upper bounds on the elements of $\mathcal{V}^{D_m}$ in terms of $\|m\|_1$ given by (12). The third part is to provide examples of situations where the extreme values occur (13). In Figure 4, a case that is extreme both in the lower and in the upper sense is illustrated.

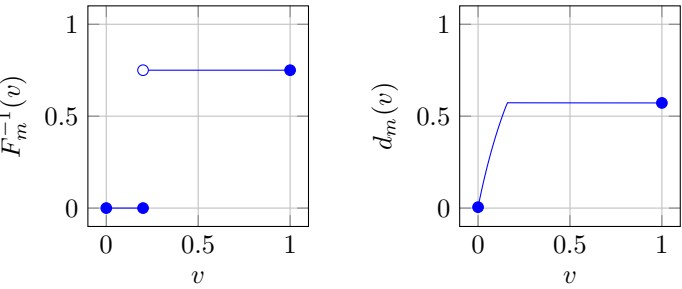

Figure 4: To the left is a particular quantile function $F_m^{-1}$ and to the right is the associated function $d_m$ given by (14). Here $F_m^{-1}(v) = (1 - \|m\|_1)(1 - \|m\|_1^2)^{-1} I_{(\|m\|_1^2, 1]}(v)$ with $\|m\|_1 = 0.4$ that satisfies $\mathcal{V}^{D_m} = [\|m\|_1^2, 1]$ and consequently is an extreme case to (13) in both the lower sense and the upper sense.

In Theorem 3, we relate the volume of the optimal segmentations of Accuracy and the optimal segmentations of Dice for a given marginal probability $m \in \mathcal{M}$.

**Theorem 3.** *For any $m \in \mathcal{M}$, $\mathcal{V}^{A_m}$ given by (7) and $\mathcal{V}^{D_m}$ given by (11) satisfy*

$$\sup \mathcal{V}^{A_m} \leq \inf \mathcal{V}^{D_m}. \tag{15}$$

The complete proof is given in the Supplementary Document and outlined as follows. First note that $D_m(s) \leq 1$ for any $s \in \mathcal{S}$ and then consider separately the cases when $D_m(s) < 1$ for all $s \in \mathcal{S}$ and when there exist some $s \in \mathcal{S}$ such that $D_m(s) = 1$. For the first case, it is obvious that $\sup_{s \in \mathcal{S}} D_m(s)/2 < 1/2$ which implies that $F_m(1/2) \leq F_m((1 - \sup_{s \in \mathcal{S}} D_m(s)/2)-)$. For the second case, we show that the volume of the optimizers are uniquely given by $\mathcal{V}^{A_m} = \mathcal{V}^{D_m} = \{F_m(1/2)\}$. In either case, (15) holds.

In Theorem 4, the set of optimal solutions to Accuracy and Dice when constrained to a specific volume is shown to coincide.

**Theorem 4.** *For any $m \in \mathcal{M}$ and $v \in [0, 1]$ the maximizers to the problems,*

$$\sup_{s \in \mathcal{S}_v} A_m(s) \quad and \quad \sup_{s \in \mathcal{S}_v} D_m(s), \tag{16}$$

*coincide and are given by $\mathcal{S}_{m,v}$.*

The complete proof is given in the Supplementary Document and is a straightforward application of Lemma 1. Of particular interest is the case when $v = \|m\|_1$, since this correspond to the situation when the metrics are maximized under the constraint that there should be no volume bias.

It follows from Theorem 1 and Theorem 2 that the optimizers to both Accuracy and Dice are of the form $\cup_{v \in \mathcal{V}} \mathcal{S}_{m,v}$, where $\mathcal{V} = [F_m(t-), F_m(t)]$ for some $t \in [0, 1]$. This type of charecterization is practical for proving properties on volume, but inconvenient for other tasks. In Theorem 5, we provide an alternative charecterization using threshold segmentations of the form $s(\omega) = I\{m(\omega) > \alpha\}$ or $s(\omega) = I\{m(\omega) \geq \alpha\}$, for some $\alpha$. Even if there exist optimal segmentations that are not necessarily of threshold type, they can always be bounded, above and below, by optimal segmentations of threshold type.

**Theorem 5.** *For any $m \in \mathcal{M}$ and $t \in (0,1]$, let $s_1(\omega) = I\{m(\omega) \geq 1-t\}$ and $s_0(\omega) > 1-t\}$. Then, $\|s_0\|_1 = F_m(t-)$, $\|s_1\|_1 = F_m(t)$ and*

$$s \in \cup_{v \in [F_m(t-), F_m(t)]} \mathcal{S}_{m,v} \iff s_0(\omega) \leq s(\omega) \leq s_1(\omega), \quad \lambda - a.e. \quad (17)$$

The complete proof is given in the Supplementary Document and outlined as follows. Note that $\|s_0\|_1 = \int_\Omega I\{\bar{m}(\omega) < t\}\lambda(d\omega) = F_m(t-)$ and $\|s_1\|_1 = \int_\Omega I\{\bar{m}(\omega) \leq t\}\lambda(d\omega) = F_m(t)$. Now, take each direction of (17) separately. For the $\Rightarrow$ part, we first show the upper bound $s(\omega) \leq s_1(\omega)$, $\lambda$-a.e. and then show the lower bound $s_0(\omega) \leq s(\omega)$, $\lambda$-a.e. For the upper bound, with $A = \{\omega : s(\omega) > s_1(\omega)\}$, we first observe that $I\{\omega \in A\} = s(\omega)\bar{s}_1(\omega)$ and then, using the definition of $\mathcal{S}_{m,v}$ we prove that

$$\lambda(A) = \int_\Omega s(\omega)\bar{s}_1(\omega)\lambda(d\omega) \leq \int_\Omega s(\omega)I\{\bar{m}(\omega) > F_m^{-1}(v)\}\lambda(d\omega) = 0. \quad (18)$$

The lower bound is similar, but slightly more involved. For the $\Rightarrow$ part, we first note that the $F_m(t-) = \|s_0\|_1 \leq \|s\|_1 \leq \|s_1\|_1 = F_m(t)$, and then show that for $v = \|s\|_1$, $s \in \mathcal{S}_{m,v}$.

In numerical applications, the continuum $\Omega$ is usually partitioned into a finite collection of voxels $\{\Omega_i\}_{i \in \mathcal{I}}$. Marginal functions are then constrained to the subset of $\mathcal{M}$ that is compatible with the voxelization in the sense that $m$ is measurable with respect to the $\sigma$-field generated by the partition. Note that if $s_1(\omega) = I\{m(\omega) \geq 1-t\}$ and $s_0(\omega) = I\{m(\omega) > 1-t\}$ for some $t \in (0,1]$, then also $s_0$ and $s_1$ are compatible with the voxelization. By Theorem 1 (Theorem 2) and Theorem 5, the segmentations with least and greatest volume that are optimal with respect to Accuracy (Dice) are compatible with the voxelization. For the metrics respectively, we denote the segmentations with the greatest volume by:

$$s^{A_m}(\omega) \doteq I\{m(\omega) \geq 1/2\}, \quad \omega \in \Omega, \quad (19)$$

$$s^{D_m}(\omega) \doteq I\{m(\omega) \geq \sup_{s \in \mathcal{S}} D_m(s)/2\}, \quad \omega \in \Omega. \quad (20)$$

Note that $s^{A_m}$ is analogous to the Bayes classifier and $s^{D_m}$ is analogous to the threshold classifier described in [18]. Both of these are trivial to compute from a given marginal function $m$ compatible to some voxelization and code for doing so is available in the Supplementary Material.

## 5 Experiments

The sharp bounds on volume in Thereom 1 and Theorem 2 implies that there exist marginal functions for which the volume of the optimal segmentations to Accuracy and Dice deviate significantly from the expected target volume. In this section we conduct experiments on marginal functions formed from real world data to compare the volume of optimal segmentations to the expected volume in practice.

For our experiments we investigate two data sets. The first data set (G) contains segmentations in the pelvic area and is part of the Gold Atlas project [25]. The data is in 3D with a resolution of $512 \times 512$ pixels per slice and consist of 19 patients with 9 different ROI's (region of interest), each of which have been delineated by 5 experts (see Figure 1 for an illustration of the segmentations associated with two different ROI's for one patient). The second data set (L) contains segmentations in the thorax area and is part of The Lung Image Database Consortium (LIDC) and Image Database Resource Initiative (IDRI) [1] and is hosted by TCIA [7]. The data is in 3D with a resolution of $512 \times 512$ pixels per slice and contains 1018 cases with lung nodules delineated by 4 experts. For each data set, ROI and patient, a marginal function $m$ is formed by taking the fraction of which each pixel has been selected by the annotators, that is, a finite sample approximation is considered. The resulting marginal functions are then used to compute the segmentations $s^{A_m}$ (19) and $s^{D_m}$ (20). For (G) we make use of the software *Plastimatch* [30] and for (L) we make use of the python package *pylidc* [12]. Details on the experiments can be found in the Supplementary Document. Code and instructions on how to reproduce the experiments can be found at https://github.com/marcus-nordstrom/optimal-solutions-to-accuracy-and-dice.

From our experiments we report the quantities $\|s^{A_m}\|_1/\|m\|_1$ and $\|s^{D_m}\|_1/\|m\|_1$, which in a relative sense describe how much the volume of the computed optimal segmentations with respect to Accuracy and Dice deviate from the expected target volume. In Figure 5 and Figure 6, these quantities are

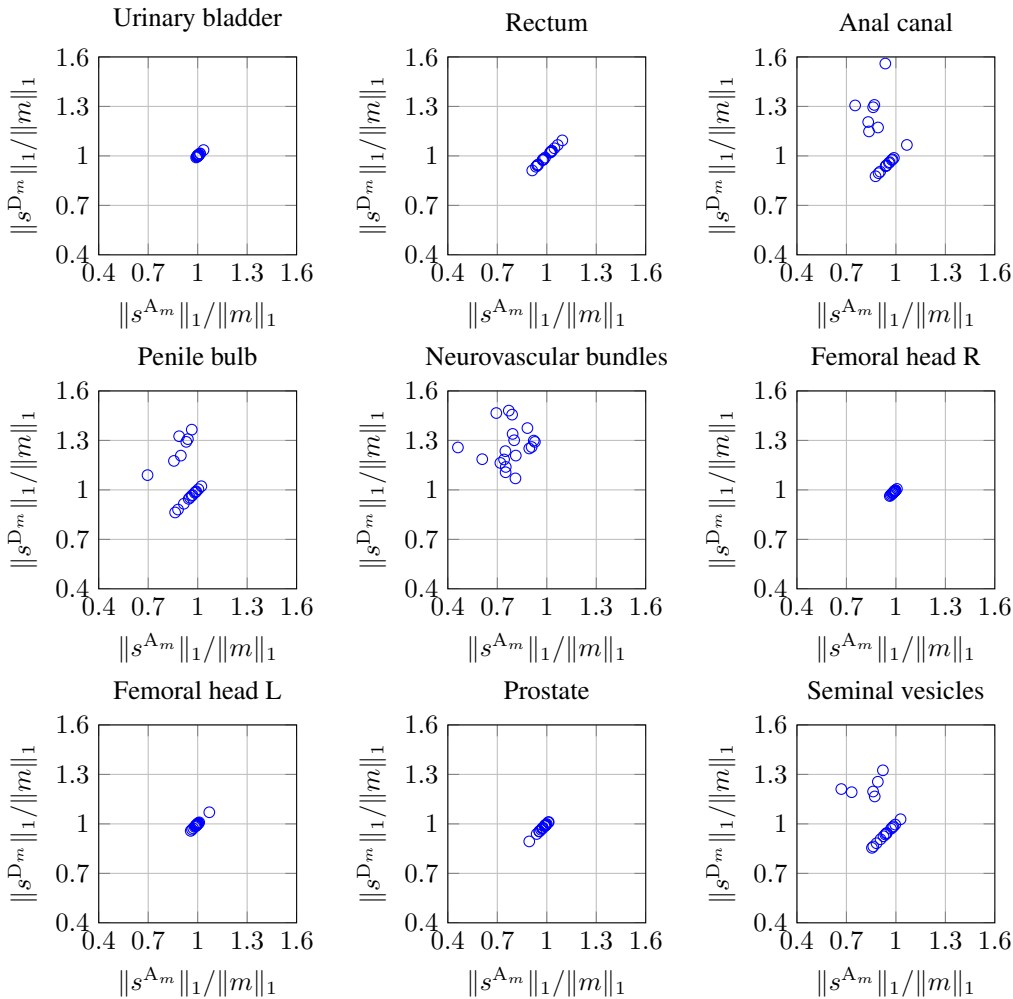

Figure 5: Results of our experiments on the pelvic data in the Gold Atlas project [25]. For each ROI and patient, the associated $m$ is formed by finite sample approximation and used to compute $s^{\mathrm{A}_m}$ as defined by (19) and $s^{\mathrm{D}_m}$ as defined by (20).

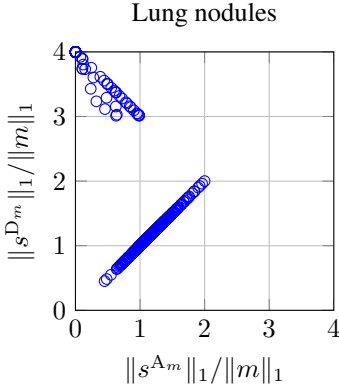

Figure 6: Results of our experiments on the lung nodules in the LIDC-IDRI data set [1]. For each patient, the associated $m$ is formed by finite sample approximation and used to compute $s^{\mathrm{A}_m}$ as defined by (19) and $s^{\mathrm{D}_m}$ as defined by (20).

Table 1: Results of our experiments on the pelvic data in the Gold Atlas project (G) [25] and the lung nodule data in LIDC-IDRI (L) [1]. For each ROI and patient, the associated $m$ is formed by finite sample approximation and used to compute $s^{\mathrm{A}m}$ as defined by (19) and $s^{\mathrm{D}m}$ as defined by (20).

| | $\|s^{\mathrm{A}m}\|_1/\|m\|_1$ | | | | $\|s^{\mathrm{D}m}\|_1/\|m\|_1$ | | | |
| ROI | Mean | Std | Min | Max | Mean | Std | Min | Max |
| --- | --- | --- | --- | --- | --- | --- | --- | --- |
| (G) Urinary bladder | 1.004 | 0.009 | 0.991 | 1.035 | 1.004 | 0.009 | 0.991 | 1.035 |
| (G) Rectum | 0.994 | 0.047 | 0.912 | 1.094 | 0.994 | 0.047 | 0.912 | 1.094 |
| (G) Anal canal | 0.916 | 0.068 | 0.753 | 1.067 | 1.075 | 0.182 | 0.877 | 1.560 |
| (G) Penile bulb | 0.929 | 0.072 | 0.696 | 1.022 | 1.065 | 0.157 | 0.863 | 1.365 |
| (G) Neurovascular b. | 0.778 | 0.110 | 0.461 | 0.928 | 1.267 | 0.115 | 1.070 | 1.481 |
| (G) Femoral head R | 0.988 | 0.011 | 0.963 | 1.006 | 0.988 | 0.011 | 0.963 | 1.006 |
| (G) Femoral head L | 0.994 | 0.022 | 0.958 | 1.070 | 0.994 | 0.022 | 0.958 | 1.070 |
| (G) Prostate | 0.978 | 0.027 | 0.894 | 1.011 | 0.978 | 0.027 | 0.894 | 1.011 |
| (G) Seminal vesicles | 0.903 | 0.085 | 0.669 | 1.028 | 1.029 | 0.142 | 0.855 | 1.325 |
| (L) Lung nodules | 1.002 | 0.362 | 0.000 | 2.000 | 1.432 | 0.893 | 0.451 | 4.000 |

illustrated for each patient and ROI in scatter plots. In Table 1, aggregated statistics of these quantities with respect to all patients are shown. By simple inspection it is clear that the volume of optimal segmentations to Accuracy and Dice often deviate significantly from the expected target volume. For (G) the extreme cases are given by some marginal function $m$ for which $\|s^{\mathrm{A}m}\|_1/\|m\|_1 \approx 0.5$ and some marginal function $m$ for which $\|s^{\mathrm{D}m}\|_1/\|m\|_1 \approx 1.5$. For (L) the extreme cases are given by some marginal function $m$ for which $\|s^{\mathrm{A}m}\|_1/\|m\|_1 \approx 0$ and some marginal function $m$ for which $\|s^{\mathrm{D}m}\|_1/\|m\|_1 \approx 4$.

## 6  Conclusion

In this work, we have theoretically investigated the optimal segmentations with respect to the performance metrics Accuracy and Dice. We have given a detailed rigorous characterization of the optimizers and upper and lower bounds on the volume of optimal segmentations. Finally, we have shown the relevance of our theoretical observations in practice by comparing the volume of optimal segmentations with respect to the performance metrics to the expected volume, on two real world data sets. We conclude that noise may cause optimal segmentations to have a volume that deviates significantly from the expected target volume and that the reason for this may be what metric is chosen, or implicitly, what metric a chosen training method targets.

**Broader impacts:**  Formalizing the evaluation process of automated segmentation methods can be done in many ways, each with its pros and cons. Even if this work can be interpreted as describing the problems with using Dice for this formalization, it can still paradoxically contribute to an unhealthy fixation of Dice as the gold standard for segmentation evaluation in medical image analysis. This in turn can lead to that medical practitioners put too much faith in models that have been shown to perform well with respect to the metric on some test data. One solution to this is to make sure that clinical practitioners using such models are educated in the problems associated with the metric.

**Limitations:**  In order for the volume bounds to be sharp in Theorem 1 and Theorem 2, we construct extreme cases of $m$. These extreme cases might only be representable approximately with step functions for a particular choice of voxelization. Consequently, the most extreme cases we can construct in a numerical setting may not be as extreme as those we have constructed in the continuous setting. However, in medical image analysis it is common to deal with voxelizations of the order of $512 \times 512 \times 100$ voxels which means that the approximation error would be negligible. Our work is also limited by the amount of experiments included. Additional numerical experiments on a wider range of data sets would give a more comprehensive picture on the impact of different noise distributions.

**Acknowledgement:**  The authors were supported by RaySearch Laboratories AB.

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
