# — Supplementary Document —
# On Image Segmentation With Noisy Labels: Characterization and Volume Properties of the Optimal Solutions to Accuracy and Dice

**Marcus Nordström** [*]
Department of Mathematics
KTH Royal Institute of Technology
Stockholm, Sweden
marcno@kth.se

**Henrik Hult**
Department of Mathematics
KTH Royal Institute of Technology
Stockholm, Sweden
hult@kth.se

**Jonas Söderberg**
Department of Machine Learning
RaySearch Laboratories
Stockholm, Sweden
jonas.soderberg@raysearchlabs.com

**Fredrik Löfman**
Department of Machine Learning
RaySearch Laboratories
Stockholm, Sweden
fredrik.lofman@raysearchlabs.com

## Contents

[*]Author is also affiliated with RaySearch Laboratories.

36th Conference on Neural Information Processing Systems (NeurIPS 2022).

# 1 Proofs

## 1.1 Generalities

A lot of our work will be devoted to analyzing cumulative distribution functions and associated quantile functions. There are several properties and identities that we will use that can be found in any standard textbook on the topic. Here is an incomplete list. Let $F(t)$, $t \in [0,1]$ be a cumulative distribution and let $F^{-1}(v) = \inf\{t : F(t) \geq v\}$ be the associated quantile function. Then the following holds.

1. $F$ is non-decreasing and right-continuous
2. $F(0-) = 0$ and $F(1) = 1$
3. $0 \leq F(t) \leq 1$, $t \in [0,1]$.
4. $F^{-1}$ is non-decreasing and left-continuous.
5. $0 \leq F^{-1}(v) \leq 1$, $v \in [0,1]$
6. $F^{-1}(F(t)) \leq t$ and $F(F^{-1}(v)) \geq v$.
7. $F^{-1}(v) \leq t$ if and only if $F(t) \leq v$.
8. if $U$ is a uniform random variable over $[0,1]$, then $F^{-1}(U)$ has the cumulative distribution function $F$

## 1.2 Proof of Lemma 1

*Proof.* Our proof consist of two parts. In the first part, we show that the class of maximizers to

$$\sup_{s \in \mathcal{S}_v} \int_\Omega s(\omega) m(\omega) \lambda(d\omega) \tag{1}$$

is given by $\mathcal{S}_{m,v}$. In the second part, we show that if $s^* \in \mathcal{S}_{m,v}$, then

$$\int_\Omega s^*(\omega) m(\omega) \lambda(d\omega) = \int_0^v (1 - F_m^{-1}(u)) du. \tag{2}$$

This together shows the statement.

**Part 1, optimality.** Let $s^* \in \mathcal{S}_{m,v}$ and $s \in \mathcal{S}_v$. Then $s^*$ and $s$ are feasible. By construction, we have that there must exist some non-negative numbers $\epsilon_1, \epsilon_2 \geq 0$ such that

$$\int_\Omega (s^*(\omega) - s(\omega)) I\{m(\omega) < 1 - F_m^{-1}(v)\} \lambda(d\omega) = -\epsilon_1, \tag{3}$$

$$\int_\Omega (s^*(\omega) - s(\omega)) I\{m(\omega) = 1 - F_m^{-1}(v)\} \lambda(d\omega) = \epsilon_1 - \epsilon_2, \tag{4}$$

$$\int_\Omega (s^*(\omega) - s(\omega)) I\{m(\omega) > 1 - F_m^{-1}(v)\} \lambda(d\omega) = \epsilon_2. \tag{5}$$

If $s \in \mathcal{S}_{m,v}$, then $\epsilon_1 = \epsilon_2 = 0$ and consequently

$$\int_\Omega (s^*(\omega) - s(\omega)) m(\omega) \lambda(d\omega) = 0. \tag{6}$$

Otherwise, if $s \in \mathcal{S}_v \setminus \mathcal{S}_{m,v}$, then at least one of the inequaities $\epsilon_1 > 0$ and $\epsilon_2 > 0$ hold. If $\epsilon_1 > 0$, it follows that

$$\int_\Omega (s^*(\omega) - s(\omega)) I\{m(\omega) < 1 - F_m^{-1}(v)\} m(\omega) \lambda(d\omega) > -\epsilon_1 (1 - F_m^{-1}(v)) \tag{7}$$

$$\int_\Omega (s^*(\omega) - s(\omega)) I\{m(\omega) = 1 - F_m^{-1}(v)\} m(\omega) \lambda(d\omega) = (\epsilon_1 - \epsilon_2)(1 - F_m^{-1}(v)) \tag{8}$$

$$\int_\Omega (s^*(\omega) - s(\omega)) I\{m(\omega) > 1 - F_m^{-1}(v)\} m(\omega) \lambda(d\omega) \geq \epsilon_2 (1 - F_m^{-1}(v)) \tag{9}$$

and similarly, if $\epsilon_2 > 0$, then

$$\int_\Omega (s^*(\omega) - s(\omega))I\{m(\omega) < 1 - F_m^{-1}\}m(\omega)\lambda(d\omega) \geq -\epsilon_1(1 - F_m^{-1}(v)) \tag{10}$$

$$\int_\Omega (s^*(\omega) - s(\omega))I\{m(\omega) = 1 - F_m^{-1}\}m(\omega)\lambda(d\omega) = (\epsilon_1 - \epsilon_2)(1 - F_m^{-1}(v)) \tag{11}$$

$$\int_\Omega (s^*(\omega) - s(\omega))I\{m(\omega) > 1 - F_m^{-1}\}m(\omega)\lambda(d\omega) > \epsilon_2(1 - F_m^{-1}(v)). \tag{12}$$

In either case, at least one inequality is strict and it follows that

$$\int_\Omega (s^*(\omega) - s(\omega))m(\omega)\lambda(d\omega) > 0. \tag{13}$$

We conclude that $\mathcal{S}_{m,v}$ is the set of optimal segmentations where the supremum is attained.

**Part 2, equality.** Let $s^* \in \mathcal{S}_{m,v}$ and introduce

$$C_1 \doteq \int_\Omega \bar{m}(\omega)I\{\bar{m}(\omega) \leq F_m^{-1}(v)\}\lambda(d\omega). \tag{14}$$

and

$$C_2 \doteq F_m^{-1}(v)(F_m(F_m^{-1}(v)) - v) \tag{15}$$

We now interpret $\bar{m}$ as a random variable on $(\Omega, \lambda)$ with cdf given by $F_m$ and then note by the quantile transform, that $F_m^{-1}(U)$ also has cdf given by $F_m$, where $U$ is uniformly distributed random variable on $[0, 1]$. It follows for $C_1$ that

$$C_1 = \mathbb{E}[F_m^{-1}(U)I\{F_m^{-1}(U) \leq F_m^{-1}(v)\}] = \mathbb{E}[F_m^{-1}(U)I\{U \leq F_m(F_m^{-1}(v))\}] \tag{16}$$

and for $C_2$ that

$$C_2 = \mathbb{E}[F_m^{-1}(v)I\{v < U \leq F_m(F_m^{-1}(v))\}] = \mathbb{E}[F_m^{-1}(U)I\{v < U \leq F_m(F_m^{-1}(v))\}] \tag{17}$$

Together with the definition of $\mathcal{S}_{m,v}$ this yields

$$\int_\Omega s^*(\omega)\bar{m}(\omega)\lambda(d\omega) = C_1 - C_2 = \mathbb{E}[F_m^{-1}(U)I\{U \leq v\}] = \int_0^v F^{-1}(u)du \tag{18}$$

Consequently,

$$\int_\Omega s^*(\omega)m(\omega)\lambda(d\omega) = v - \left(\int_\Omega s^*(\omega)\lambda(d\omega) - \int_\Omega s(\omega)m(\omega)\lambda(d\omega)\right) \tag{19}$$

$$= v - \int_\Omega s^*(\omega)\bar{m}(\omega)\lambda(d\omega) \tag{20}$$

$$= v - \int_0^v F^{-1}(u)du \tag{21}$$

$$= \int_0^v (1 - F^{-1}(u))du \tag{22}$$

This completes the proof.

$\square$

## 1.3 Proof of Theorem 1

*Proof.* Consider the function

$$a_m(v) \doteq v + 1 - \|m\|_1 - 2\int_0^v F_m^{-1}(u)du, \quad v \in [0, 1]. \tag{23}$$

and note that by Lemma 1

$$\sup_{s \in \mathcal{S}} A_m(s) = \sup_{s \in \mathcal{S}} \int_\Omega [s(\omega)m(\omega) + \bar{s}(\omega)\bar{m}(\omega)]\lambda(d\omega) \tag{24}$$

$$= \sup_{v \in [0,1]} \sup_{s \in \mathcal{S}_v} \int_\Omega [2s(\omega)m(\omega) + 1 - s(\omega) - m(\omega)]\lambda(d\omega) \tag{25}$$

$$= \sup_{v \in [0,1]} \left[ 1 - \|m\|_1 - v + 2 \sup_{s \in \mathcal{S}_v} \int_\Omega s(\omega)m(\omega)\lambda(d\omega) \right] \tag{26}$$

$$= \sup_{v \in [0,1]} \left[ 1 - \|m\|_1 - v + 2 \int_0^v (1 - F_m^{-1}(u))du \right] \tag{27}$$

$$= \sup_{v \in [0,1]} \left[ v + 1 - \|m\|_1 - 2 \int_0^v F_m^{-1}(u)du \right] \tag{28}$$

$$= \sup_{v \in [0,1]} a_m(v) \tag{29}$$

$$= a_m(v^*), \quad v^* \in \mathcal{V}^* \tag{30}$$

where the supremum of $A_m$ is attained for segmentations in $\cup_{v^* \in \mathcal{V}^*} \mathcal{S}_{m,v^*}$. Moreover, the function $a_m$ is differentiable a.e., with derivative given by

$$\partial_v a_m(v) = 1 - 2F_m^{-1}(v).$$

**Part 1, characterization.** Since $F_m^{-1}$ is non-decreasing it follows that $\partial_v a_m$ is positive for $F_m^{-1}(v) < \frac{1}{2}$ and negative for $F_m^{-1}(v) > \frac{1}{2}$. Therefore, the maximum of $a_m$ is attained for $v$ such that $F_m^{-1}(v) = \frac{1}{2}$. That is, the maximum is attained for $v \in (F_m(\frac{1}{2}-), F_m(\frac{1}{2})]$. For $v$ in this set,

$$a_m(v) = v + 1 - \|m\|_1 - 2 \left( (v - F_m(\tfrac{1}{2}-))\frac{1}{2} + \int_0^{F_m(\frac{1}{2}-)} F_m^{-1}(u)du \right) \tag{31}$$

$$= F_m(\tfrac{1}{2}-) + 1 - \|m\|_1 - \int_0^{F_m(\frac{1}{2}-)} F_m^{-1}(u)du. \tag{32}$$

**Part 2, bounds.** We now prove the lower bound for $F_m(\frac{1}{2}-)$ and the upper bound for $F_m(\frac{1}{2})$. For this, first note that if we interpret $\bar{m}$ as a random variable on $(\Omega, \lambda)$, with cdf given by $F_m$, and then note by the quantile transform, that $F_m^{-1}(U)$ also has cdf given by $F_m$ where $U$ is a uniformly distributed random variable on $[0,1]$, then

$$\int_0^1 (1 - F_m^{-1}(u))du = 1 - \mathbb{E}[F^{-1}(U)] = 1 - \int_\Omega \bar{m}(\omega)\lambda(d\omega) = \|m\|_1. \tag{33}$$

Now, for the lower bound, note that, for $v \in [0,1]$,

$$1 - \|m\|_1 = \int_0^1 F_m^{-1}(u)du \geq (1-v)F_m^{-1}(v), \tag{34}$$

which implies that

$$F_m^{-1}(v) \leq \frac{1 - \|m\|_1}{1 - v}. \tag{35}$$

Consequently, for $\|m\|_1 \geq 1/2$,

$$F_m^{-1}(v) \leq \frac{1 - \|m\|_1}{1 - v} < \frac{1}{2}, \quad 0 \leq v < 2\|m\|_1 - 1 \tag{36}$$

and

$$F_m(\tfrac{1}{2}-) \geq \lim_{v \uparrow 1 - 2(1 - \|m\|_1)} F_m(F_m^{-1}(v)) \geq 1 - 2(1 - \|m\|_1) = 2\|m\|_1 - 1. \tag{37}$$

For $\|m\|_1 < 1/2$, clearly $F_m(\frac{1}{2}-) \geq 0$. The cases $\|m\|_1 \geq 1/2$ and $\|m\|_1 < 1/2$ together means that

$$F(1/2-) \geq \max\{2\|m\|_1 - 1, 0\}. \tag{38}$$

For the upper bound, note that, for $v \in [0,1]$,

$$1 - \|m\|_1 = \int_0^1 F_m^{-1}(u)du \leq vF_m^{-1}(v) + 1 - v, \tag{39}$$

which implies that

$$F_m^{-1}(v) \geq 1 - \frac{\|m\|_1}{v}. \tag{40}$$

Consequently, with $v = F_m(\frac{1}{2})$

$$1 - \frac{\|m\|_1}{F_m(\frac{1}{2})} \leq F_m^{-1}(F(\frac{1}{2})) \leq \frac{1}{2}, \tag{41}$$

and it follows that $F_m(\frac{1}{2}) \leq 2\|m\|_1$. For $\|m\|_1 \geq \frac{1}{2}$ we clearly have that $F_m(\frac{1}{2}) \leq 1$. The cases $\|m\|_1 < 1/2$ and $\|m\|_1 \geq 1/2$ together means that

$$F(1/2) \leq \min\{2\|m\|_1, 1\}. \tag{42}$$

**Part 3, sharpness.** Recall that the domain $\Omega = [0,1]^n$ for some $n \geq 1$. We use $(\omega_1, \ldots, \omega_n) = \omega \in \Omega$ to denote the components.

To see that the lower bound is sharp, take $v \geq \frac{1}{2}$ and $m_0 \in \mathcal{M}$ given by

$$m_0(\omega) = I_{[0,2v-1]}(\omega_1) + \frac{1}{2}I_{(2v-1,1]}(\omega_1), \quad \omega \in \Omega. \tag{43}$$

Note that

$$\|m_0\|_1 = (2v - 1) + \frac{1}{2}(1 - (2v - 1)) = v. \tag{44}$$

Consequently,

$$F_{m_0}(t) = (2v - 1)I_{[0,\frac{1}{2})}(t) + I_{[\frac{1}{2},1]}(t), \quad t \in [0,1], \tag{45}$$

which means that $F_{m_0}(\frac{1}{2}-) = 2v - 1$. Now instead take $v < 1/2$ and $m_0 \in \mathcal{M}$ given by

$$m_0(\omega) = vI_{[0,1]}(\omega_1), \quad \omega \in \Omega. \tag{46}$$

Note that

$$\|m_0\|_1 = v(1 - 0) = v. \tag{47}$$

Furthermore,

$$F_{m_0}(t) = I_{[1-v,1]}(t), \quad t \in [0,1], \tag{48}$$

which means that $F_{m_0}(\frac{1}{2}-) = 0$. Together, these cases say that for any $v \in [0,1]$, the $m_0$ formed by taking

$$m_0(\omega) = \begin{cases} I_{[0,2v-1]}(\omega_1) + \frac{1}{2}I_{(2v-1,1]}(\omega_1), & \text{if } v \geq 1/2, \\ vI_{[0,1]}(\omega_1), & \text{if } v < 1/2, \end{cases} \quad \omega \in \Omega, \tag{49}$$

satisfies $\|m_0\|_1 = v$ and

$$F_{m_0}(\frac{1}{2}-) = \max\{2\|m_0\|_1 - 1, 0\}. \tag{50}$$

To see that the upper bound is sharp, take $v < \frac{1}{2}$ and $m_1 \in \mathcal{M}$ such that

$$m_1(\omega) = \frac{1}{2}I_{[0,2v]}(\omega_1), \quad \omega \in \Omega. \tag{51}$$

Note that

$$\|m_1\|_1 = \frac{1}{2}(2v - 0) = v. \tag{52}$$

Furthermore,

$$F_{m_1}(t) = 2v I_{[\frac{1}{2},1)}(t) + I_{\{1\}}(t), \quad t \in [0,1], \tag{53}$$

which means that $F_{m_1}(\frac{1}{2}) = 2v$. Now instead take $v \geq 1/2$ and $m_1 \in \mathcal{M}$ given by

$$m_1(\omega) = v, \quad \omega \in \Omega. \tag{54}$$

Note that

$$\|m_1\|_1 = v. \tag{55}$$

Furthermore

$$F_{m_1}(t) = I_{[1-v,1]}(t), \quad t \in [0,1], \tag{56}$$

which means that $F_{m_1}(\frac{1}{2}) = 1$. Together, these cases say that for any $v \in [0,1]$, the $m_1$ formed by taking

$$m_1(\omega) = \begin{cases} \frac{1}{2} I_{[0,2v]}(\omega_1), & \text{if } v < 1/2 \\ v, & \text{if } v \geq 1/2, \end{cases} \quad \omega \in \Omega, \tag{57}$$

satisfies $\|m_1\|_1 = v$ and

$$F_{m_1}\left(\frac{1}{2}\right) = \min\{2\|m_1\|_1, 1\}. \tag{58}$$

This completes the proof. □

## 1.4  Proof of Theorem 2

*Proof.* Consider the function

$$d_m(v) \doteq \frac{2 \int_0^v 1 - F_m^{-1}(u)du}{\|m\|_1 + v}, \quad v \in [0,1]. \tag{59}$$

and note that by Lemma 1

$$\sup_{s \in \mathcal{S}} D(s) = \sup_{v \in [0,1]} \frac{\sup_{s \in \mathcal{S}_v} 2 \int_\Omega s(\omega)m(\omega)\lambda(d\omega)}{\|m\|_1 + v} \tag{60}$$

$$= \sup_{v \in [0,1]} \frac{2 \int_0^v 1 - F_m^{-1}(u)du}{\|m\|_1 + v} \tag{61}$$

$$= \sup_{v \in [0,1]} d_m(v) \tag{62}$$

$$= d_m(v^*), \quad v^* \in \mathcal{V}^*, \tag{63}$$

where the supremum of $D_m$ is attained for segmentations in $\cup_{v^* \in \mathcal{V}^*} \mathcal{S}_{m,v^*}$. Note that $d_m$ is continuous and hence attains its maximum on the compact set $[0,1]$. By continuity, the set $\mathcal{V}^* \subset [0,1]$ where the maximum is attained is closed and therefore compact. Moreover, $d_m$ is differentiable a.e. with derivative given by

$$\partial_v d_m(v) = \frac{2(1 - F_m^{-1}(v)) - d_m(v)}{\|m\|_1 + v}. \tag{64}$$

Consider the function $\delta$ given by

$$\delta(v) = \|m\|_1 + \int_0^v F_m^{-1}(u)du - (\|m\|_1 + v)F_m^{-1}(v), \quad v \in [0,1]. \tag{65}$$

The interest in the function $\delta$ comes from the fact that it has the same sign as $\partial_v d_m$ but is somewhat easier to work with. Indeed, note that $\delta(v) = \frac{(\|m\|_1 + v)^2}{2} \partial_v d_m(v)$ and hence, $\partial_v d_m(v) > 0$ if and only if $\delta(v) > 0$, and similarly, $\partial_v d_m(v) < 0$ if and only if $\delta(v) < 0$.

**Part 1, characterization.** Note first that $\delta$ is left-continuous and non-increasing on $[0, 1]$. That $\delta$ is left-continuous follows by construction since $F_m^{-1}$ is left continuous. That $\delta$ is non-increasing on $[0, 1]$ follows since $F_m^{-1}$ is non-decreasing and consequently that for $0 \leq v_0 < v_1 \leq 1$

$$\delta(v_1) - \delta(v_0) = \|m\|_1 (F_m^{-1}(v_0) - F_m^{-1}(v_1)) + (v_0 F_m^{-1}(v_0) - v_1 F_m^{-1}(v_1)) - \int_{v_0}^{v_1} F_m^{-1}(u) du \tag{66}$$

$$\leq \|m\|_1 \cdot 0 + 0 + 0 \tag{67}$$

$$\leq 0. \tag{68}$$

Using these properties, it follows that there are three possibilities.

(i) $\delta(v) > 0$ for all $v \in [0, 1)$, in which case $d_m$ is strictly increasing on $[0, 1)$ and attains its maximum at $v^* = 1$. In this case $\mathcal{V}^* = \{1\}$.

(ii) There is a half-open interval $(a, b] \subset [0, 1]$ where $\delta = 0$. By continuity $d_m$ is constant on the closed interval $\mathcal{V}^* = [a, b]$ and attains its maximum on $\mathcal{V}^*$.

(iii) There is a unique point $v^* \in [0, 1]$ such that $\delta(v) > 0$ for $v < v^*$ and $\delta(v) < 0$, for $v > v^*$. In this case $\mathcal{V}^* = \{v^*\}$ and $d$ attains its maximum at $v^*$.

In each case $\mathcal{V}^*$ is a closed subinterval of $[0, 1]$. Now we prove that it is exactly the interval $\mathcal{V}^* = [F_m((1 - d_m^*/2)-), F_m(1 - d_m^*/2)]$ where $d_m^* = d_m(v^*)$ for $v^* \in \mathcal{V}^*$ and $F_m((1 - d_m^*/2)-) = \lim_{t \uparrow 1 - d_m^*/2} F_m(t)$ denotes the left limit. To this end, consider the three cases (i), (ii), and (iii) separately.

In case (i) $\mathcal{V}^* = \{1\}$, so it is sufficient to show that $F_m((1 - \frac{d_m^*}{2})-) = 1$. Recall that $\delta(v) > 0$ for $v < 1$ and $\delta$ is left-continuous it follows that $\delta(1) \geq 0$ and by definition of $\delta$ that

$$F_m^{-1}(1) \leq 1 - \frac{d_m(1)}{2} = 1 - \frac{d_m^*}{2}. \tag{69}$$

There are two cases to consider. If $F_m^{-1}(1) = 1 - \frac{d_m^*}{2}$, then $\delta(1) = 0$ and we claim that $F_m^{-1}(v) < F_m^{-1}(1)$, for $v < 1$. Indeed, otherwise $F_m^{-1}(v_0) = F_m^{-1}(1)$ for some $v_0 < 1$, which implies that $F_m^{-1}(v) = F_m^{-1}(1)$ for all $v \in [v_0, 1]$ and, by (66), $\delta(v)$ is constant on $[v_0, 1]$. But since by assumption $\delta(v) > 1$ for $v \in [v_0, 1)$, this means that $\delta(1) > 0$ which is a contradiction. In the other case, if $F_m^{-1}(1) < 1 - \frac{d_m^*}{2}$, then $F_m^{-1}(v) < 1 - \frac{d_m^*}{2}$, since $F_m^{-1}$ is non-decreasing. In both cases it holds that $F_m^{-1}(v) < 1 - \frac{d_m^*}{2}$, for $v < 1$, and consequently,

$$1 = \lim_{v \uparrow 1} v \leq \lim_{v \uparrow 1} F_m(F_m^{-1}(v)) \leq F_m((1 - \frac{d_m^*}{2})-). \tag{70}$$

This completes the proof in case (i).

In case (ii), with $\mathcal{V}^* = [a, b]$, $a < b$, we show first that $a = F_m((1 - \frac{d_m^*}{2})-)$ by showing the inequalities $a \leq F_m((1 - \frac{d_m^*}{2})-)$ and $a \geq F_m((1 - \frac{d_m^*}{2})-)$. The first inequality is similar to the proof of case (i) above. For $v < a$, $\delta(v) > 0$, and left-continuity of $\delta$ implies that $\delta(a) \geq 0$ and it follows that

$$F_m^{-1}(a) \leq 1 - \frac{d_m(a)}{2} = 1 - \frac{d_m^*}{2}. \tag{71}$$

There are two cases to consider. If $F_m^{-1}(a) = 1 - \frac{d_m^*}{2}$, then $\delta(a) = 0$ and we claim that $F_m^{-1}(v) < F_m^{-1}(a)$, for $v < a$. Indeed, otherwise $F_m^{-1}(v_0) = F_m^{-1}(a)$ for some $v_0 < a$, which implies that $F_m^{-1}(v) = F_m^{-1}(a)$ for all $v \in [v_0, a]$ and, by (66), $\delta(v)$ is constant on $[v_0, a]$. But since by assumption $\delta(v) > 0$ for $v \in [v_0, a)$, this means that $\delta(a) > 0$ which is a contradiction. In the other case, if $F_m^{-1}(a) < 1 - \frac{d_m^*}{2}$, then $F_m^{-1}(v) < 1 - \frac{d_m^*}{2}$, since $F_m^{-1}$ is non-decreasing. In both cases it holds that $F_m^{-1}(v) < 1 - \frac{d_m^*}{2}$, for $v < a$, and consequently,

$$a = \lim_{v \uparrow a} v \leq \lim_{v \uparrow a} F_m(F_m^{-1}(v)) \leq F_m((1 - \frac{d_m^*}{2})-). \tag{72}$$

To show the reverse inequality $a \geq F_m((1 - \frac{d_m^*}{2})-)$, note that for $t < 1 - \frac{d_m^*}{2}$ and $v < F_m(t)$, it holds that

$$F_m^{-1}(v) \leq t < 1 - \frac{d_m^*}{2} \leq 1 - \frac{d_m(v)}{2}, \tag{73}$$

which implies that $\delta(v) > 0$. Consequently, $v \leq a$ and we conclude that $F_m((1 - \frac{d_m^*}{2})-) \leq a$.

To complete the proof in case (ii) it remains to show that $F_m((1 - \frac{d_m^*}{2})) = b$. For $v \in (a, b]$ it holds that $\delta(v) = 0$ and it follows that $F_m^{-1}(v) = 1 - \frac{d_m^*}{2}$. Therefore,

$$F_m(1 - \frac{d_m^*}{2}) = F_m(F_m^{-1}(v)) \geq v. \tag{74}$$

By taking the limit as $v \uparrow b$ we conclude that $b \leq F_m(1 - \frac{d_m^*}{2})$. For the reverse inequality, since

$$F_m^{-1}(F_m(1 - \frac{d_m^*}{2})) \leq 1 - \frac{d_m^*}{2}, \tag{75}$$

it follows that $\delta(F_m(1 - \frac{d_m^*}{2})) \geq 0$ and we conclude that $F_m(1 - \frac{d_m^*}{2}) \leq b$.

In case (iii), with $\mathcal{V}^* = \{v^*\}$, it is sufficient to prove that

$$v^* \leq F_m((1 - \frac{d_m^*}{2})-) \leq F_m(1 - \frac{d_m^*}{2}) \leq v^*. \tag{76}$$

The proof of $v^* \leq F_m((1 - \frac{d_m^*}{2})-)$ is identical to the proof of $a \leq F_m((1 - \frac{d_m^*}{2})-)$ in (ii) and is therefore omitted. To prove $v^* \geq F_m((1 - \frac{d_m^*}{2}))$, recall that $\delta(v) > 0$ for $v < v^*$, $\delta(v^*) \geq 0$, by left-continuity of $\delta$, and $\delta(v) < 0$, $v > v^*$. Since

$$F_m^{-1}(F_m(1 - \frac{d_m^*}{2})) \leq 1 - \frac{d_m^*}{2}, \tag{77}$$

it follows that $\delta(F_m(1 - \frac{d_m^*}{2})) \geq 0$ and we conclude that $F_m(1 - \frac{d_m^*}{2}) \leq v^*$.

This completes the proof of $\mathcal{V}^* = [F_m((1 - d_m^*/2)-), F_m(1 - d_m^*/2)]$.

**Part 2, bounds.** We now show that $\mathcal{V}^* \in [\|m\|_1^2, 1]$. Note that the upper bound is the upper bound of the feasibility set $v \in [0, 1]$ so it is already done. For the lower bound, note that

$$1 - \|m\|_1 = \int_0^1 F_m^{-1}(u)du \geq \int_{1-v}^1 F_m^{-1}(u)du \geq (1 - v)F_m^{-1}(v), \tag{78}$$

which implies that

$$F_m^{-1}(v) \leq \frac{1 - \|m\|_1}{1 - v}, \quad v \in [0, 1]. \tag{79}$$

Therefore,

$$\delta(v) \geq \|m\|_1(1 - F_m^{-1}(v)) - vF_m^{-1}(v) \geq \|m\|_1 - (\|m\|_1 + v)\frac{1 - \|m\|_1}{1 - v} = \frac{\|m\|_1^2 - v}{1 - v}. \tag{80}$$

We conclude that $\delta(v) > 0$ for $v < \|m\|_1^2$, which completes the proof.

**Part 3, sharpness.** Recall that the domain $\Omega = [0, 1]^n$ for some $n \geq 1$. We use $(\omega_1, \dots, \omega_n) = \omega \in \Omega$ to denote the components.

To prove that the bounds are sharp, it is enough to consider one example. Take $v' \in (0, 1]$ and $m \in \mathcal{M}$ such that

$$m(\omega) = I_{[0,v'^2]}(\omega_1) + \frac{v'}{1 + v'}I_{(v'^2,1]}(\omega_1), \quad \omega \in \Omega. \tag{81}$$

Note that

$$\|m\|_1 = (v'^2 - 0) + \frac{v'}{1 + v'}(1 - v'^2) = v'. \tag{82}$$

Furthermore,

$$F_m(t) = v'^2 I_{[0, \frac{1}{1+v'})}(t) + I_{[\frac{1}{1+v'},1]}(t), \quad t \in [0,1], \tag{83}$$

or equivelently,

$$F_m(t) = \|m\|_1^2 I_{[0, \frac{1}{1+\|m\|_1})}(t) + I_{[\frac{1}{1+\|m\|_1},1]}(t), \quad t \in [0,1], \tag{84}$$

which means that

$$F_m^{-1}(v) = \frac{1}{1+\|m\|_1} I_{(\|m\|_1^2,1]}(v), \quad v \in [0,1]. \tag{85}$$

Then for $v \in [0,1]$

$$\delta(v) = \|m\|_1 + \int_0^v F_m^{-1}(u)du - (\|m\|_1 + v)F_m^{-1}(v) \tag{86}$$

$$= \|m\|_1 + \int_0^v \frac{1}{1+\|m\|_1} I_{(\|m\|_1^2,1]}(u)du - (\|m\|_1 + v)\frac{1}{1+m} I_{(\|m\|_1^2,1]}(v) \tag{87}$$

$$= \|m\|_1 + \frac{1}{1+\|m\|_1}(v - \|m\|_1^2)I_{(\|m\|_1^2,1]}(v) - \frac{\|m\|_1 + v}{1+\|m\|_1} I_{(\|m\|_1^2,1]}(v) \tag{88}$$

$$= \|m\|_1 - \|m\|_1 I_{(\|m\|_1^2,1]} \tag{89}$$

$$= \|m\|_1 I_{[0,\|m\|_1^2]} \tag{90}$$

Hence, $\delta(v) = 0$ for $v > \|m\|_1^2$ and $d$ attains its maximum on $\mathcal{V}^* = [\|m\|_1^2, 1]$. Since we we can choose any $v' \in (0,1]$ so that $\|m\|_1 = v'$, this completes the proof.

$\square$

## 1.5  Proof of Theorem 3

*Proof.* If we interpret $\bar{m}$ as a random variable on $(\Omega, \lambda)$, with cdf given by $F_m$, and then note by the quantile transform, that $F_m^{-1}(U)$ also has cdf given by $F_m$ where $U$ is a uniformly distributed random variable on $[0,1]$, it follows that

$$\int_0^1 (1 - F_m^{-1}(u))du = 1 - \mathbb{E}[F^{-1}(U)] = 1 - \int_\Omega \bar{m}(\omega)\lambda(d\omega) = \|m\|_1. \tag{91}$$

Now, as in Theorem 2, consider the function

$$d_m(v) = \frac{2\int_0^v (1 - F_m^{-1}(u))du}{\|m\|_1 + v}, \quad v \in [0,1]. \tag{92}$$

and note that by Lemma 1

$$\sup_{s \in \mathcal{S}} \mathrm{D}_m(s) = \sup_{v \in [0,1]} \frac{\sup_{s \in \mathcal{S}_v} 2\int_\Omega s(\omega)m(\omega)\lambda(d\omega)}{\|m\|_1 + v} \tag{93}$$

$$= \sup_{v \in [0,1]} \frac{2\int_0^v (1 - F_m^{-1}(u))du}{\|m\|_1 + v} \tag{94}$$

$$= \sup_{v \in [0,1]} d_m(v) \tag{95}$$

$$\doteq d_m^*. \tag{96}$$

This means

$$d_m(v) = \frac{2\int_0^v (1 - F_m^{-1}(u)dv)}{\|m\|_1 + v} \leq \frac{\int_0^1 (1 - F_m^{-1}(u)dv) + \int_0^v 1dv}{\|m\|_1 + v} = 1, \quad v \in [0,1], \tag{97}$$

and consequently that $d_m^* \leq 1$. We now show that

$$\sup \mathcal{V}^{\mathrm{A}_m} = F_m(1/2) \leq F_m((1 - d_m^*/2)-) = \inf \mathcal{V}^{\mathrm{D}_m} \tag{98}$$

separately for the cases $d_m^* < 1$ and $d_m^* = 1$. First, assume that $d_m^* < 1$. Then there exist some $\epsilon > 0$ such that $d_m^* < 1 - 2\epsilon$, hence

$$1 - d_m^*/2 > 1 - (1 - 2\epsilon)/2 = 1/2 + \epsilon, \tag{99}$$

and since $F_m$ is non-decreasing

$$F_m((1 - d_m^*/2)-) \geq F_m((1/2 + \epsilon)-) \geq F_m(1/2). \tag{100}$$

Secondly, assume that $d_m^* = 1$. Then there must exist some $v$ such that $d_m(v) = 1$, or equivalently

$$\int_0^v (1 - F_m^{-1}(u))du = \frac{1}{2}(\|m\|_1 + v). \tag{101}$$

If $v < \|m\|_1$, then

$$v = \int_0^v 1 dv \geq \int_0^v (1 - F_m^{-1}(u))du = \frac{1}{2}(\|m\|_1 + v) > v \tag{102}$$

which is a contradiction and if $v > |m|$ then

$$\|m\|_1 = \int_0^1 (1 - F_m^{-1}(u))du \geq \int_0^v (1 - F_m^{-1}(u))du = \frac{1}{2}(\|m\|_1 + v) > \|m\|_1 \tag{103}$$

which also is a contradiction. Consequently, we must have that $v = \|m\|_1$. This in turn means that $F_m^{-1}$ needs to satisfy

$$\int_0^{\|m\|_1} (1 - F_m^{-1}(u))du = \|m\|_1. \tag{104}$$

which can only be the case if

$$F_m^{-1}(v) = I_{(\|m\|_1, 1]}(v), \quad v \in [0, 1]. \tag{105}$$

This means that

$$F_m(t) = \|m\|_1 I_{[0,1)}(t) + I_{\{1\}}(t). \tag{106}$$

and finally that

$$F_m((1 - d_m^*/2)-) = F_m(1/2) = \|m\|_1. \tag{107}$$

This completes the proof. $\qquad\square$

## 1.6 Proof of Theorem 4

*Proof.* By Lemma 1, the maximizers to

$$\sup_{s \in \mathcal{S}_v} A_m(s) = \sup_{s \in \mathcal{S}_v} \int_\Omega [s(\omega)m(\omega) + \bar{s}(\omega)\bar{m}(\omega)]\lambda(d\omega) \tag{108}$$

$$= 1 - \|m\|_1 - v + 2 \sup_{s \in \mathcal{S}_v} \int_\Omega s(\omega)m(\omega)\lambda(d\omega) \tag{109}$$

are given by $\mathcal{S}_{m,v}$, and the maximizers to

$$\sup_{s \in \mathcal{S}_v} D_m(s) = \frac{2 \int_\Omega s(\omega)m(\omega)\lambda(d\omega)}{\|m\|_1 + \|s\|_1} \tag{110}$$

$$= \frac{2}{\|m\|_1 + v} \sup_{s \in \mathcal{S}_v} \int_\Omega s(\omega)m(\omega)\lambda(d\omega) \tag{111}$$

are given by $\mathcal{S}_{m,v}$. This completes the proof. $\qquad\square$

## 1.7 Proof of Theorem 5

*Proof.* First note that $\|s_0\|_1 = \int_\Omega I\{\bar{m}(\omega) < t\}\lambda(d\omega) = F_m(t-)$ and $\|s_1\|_1 = \int_\Omega I\{\bar{m}(\omega) \leq t\}\lambda(d\omega) = F_m(t)$. We now prove the if and only if cases separately.

**Part 1, $\Rightarrow$.** Take any $s \in \cup_{v \in [F(t-), F(t)]} \mathcal{S}_{m,v}$. To prove the upper bound $s(\omega) \leq s_1(\omega)$, $\lambda$-a.e., let $A = \{\omega : s(\omega) > s_1(\omega)\}$. Since both $s$ and $s_1$ are binary it follows that

$$I\{\omega \in A\} = s(\omega)\bar{s}_1(\omega), \quad \omega \in \Omega. \tag{112}$$

By definition of $\mathcal{S}_{m,v}$ and, since $v \leq F_m(t)$ implies that $F_m^{-1}(v) \leq F_m^{-1}(F_m(t)) \leq t$, it follows that

$$0 = \int_\Omega s(\omega) I\{\bar{m}(\omega) > F_m^{-1}(v)\} \lambda(d\omega) \geq \int_\Omega s(\omega) I\{\bar{m}(\omega) > t\} \lambda(d\omega) \geq 0. \tag{113}$$

That is,

$$\int_\Omega s(\omega) I\{\bar{m}(\omega) > t\} \lambda(d\omega) = 0. \tag{114}$$

Consequently,

$$\lambda(A) = \int_\Omega s(\omega)\bar{s}_1(\omega)\lambda(d\omega) = \int_\Omega s(\omega) I\{\bar{m}(\omega) > t\}\lambda(d\omega) = 0. \tag{115}$$

To prove the lower bound $s_0(\omega) \leq s(\omega)$, $\lambda$-a.e., let $B = \{\omega : s_0(\omega) > s(\omega)\}$. Since $s_0$ and $s$ are binary, it follows that

$$I\{\omega \in B\} = s_0(\omega)\bar{s}(\omega) = \bar{s}(\omega) I\{\bar{m}(\omega) < t\}, \quad \omega \in \Omega. \tag{116}$$

Therefore, it is sufficient to show that

$$\int_\Omega \bar{s}(\omega) I\{\bar{m}(\omega) < t\}\lambda(d\omega) = 0. \tag{117}$$

We know from the definition of $\mathcal{S}_{m,v}$ that

$$\int_\Omega \bar{s}(\omega) I\{\bar{m}(\omega) < F_m^{-1}(v)\}\lambda(d\omega) = 0. \tag{118}$$

Consequently, with $v \in [F_m(t-), F_m(t)]$ it follows that $F_m^{-1}(v) \leq t$ and

$$0 \leq \int_\Omega \bar{s}(\omega) I\{\bar{m}(\omega) < t\}\lambda(d\omega) \tag{119}$$

$$= \underbrace{\int_\Omega \bar{s}(\omega) I\{\bar{m}(\omega) < F_m^{-1}(v)\}\lambda(d\omega)}_{=0} + \int_\Omega \bar{s}(\omega) I\{F_m^{-1}(v) \leq \bar{m}(\omega) < t\}\lambda(d\omega) \tag{120}$$

$$\leq \int_\Omega I\{F_m^{-1}(v) \leq \bar{m}(\omega) < t\}\lambda(d\omega) \tag{121}$$

$$= F_m(t-) - F_m(F_m^{-1}(v)) \tag{122}$$

$$\leq F_m(t-) - v \tag{123}$$

$$\leq 0. \tag{124}$$

We conclude that (117) holds.

**Part 2, $\Leftarrow$.** Take any $s \in \mathcal{S}$ such that $s_0(\omega) \leq s(\omega) \leq s_1(\omega)$ $\lambda$-a.e. Since,

$$F_m(t-) = \|s_0\|_1 \leq \|s\|_1 \leq \|s_1\|_1 = F(t), \tag{125}$$

it follows that $s \in \mathcal{S}_v$, with $v \in [F_m(t-), F_m(t)]$. Furthermore, with $v = \|s\|_1$,

$$0 \leq \int_\Omega s(\omega) I\{m(\omega) < 1 - F_m^{-1}(v)\}\lambda(d\omega) \tag{126}$$

$$= \int_\Omega s(\omega) I\{\bar{m}(\omega) > F_m^{-1}(v)\}\lambda(d\omega) \tag{127}$$

$$\leq \int_\Omega s(\omega) I\{\bar{m}(\omega) > F_m^{-1}(F_m(t-))\}\lambda(d\omega) \tag{128}$$

$$\leq \int_\Omega s_1(\omega) I\{\bar{m}(\omega) > F_m^{-1}(F_m(t-))\}\lambda(d\omega) \tag{129}$$

$$= \int_\Omega I\{F_m^{-1}(F_m(t-)) < \bar{m}(\omega) \leq t\}\lambda(d\omega) \tag{130}$$

$$= F_m(t) - F_m(F_m^{-1}(F_m(t-))) = 0, \tag{131}$$

where the last equality follows since $F_m$ is a cumulative distribution function. Similarly,

$$0 \leq \int_\Omega \bar{s}(\omega) I\{m(\omega) > 1 - F_m^{-1}(v)\} \lambda(d\omega) \tag{132}$$

$$= \int_\Omega (1 - s(\omega)) I\{\bar{m}(\omega) < F_m^{-1}(v)\} \lambda(d\omega) \tag{133}$$

$$\leq \int_\Omega (1 - s(\omega)) I\{\bar{m}(\omega) < F_m^{-1}(F_m(t))\} \lambda(d\omega) \tag{134}$$

$$\leq \int_\Omega (1 - s_0(\omega)) I\{\bar{m}(\omega) < F_m^{-1}(F_m(t))\} \lambda(d\omega) \tag{135}$$

$$= \int_\Omega I\{t \leq \bar{m}(\omega) < F_m^{-1}(F_m(t))\} \lambda(d\omega) \tag{136}$$

$$= F_m(F_m^{-1}(F_m(t))-) - F_m(t-) \leq 0. \tag{137}$$

We conclude that any segmentation $s \in \mathcal{S}$ satisfying $s_0(\omega) \leq s(\omega) \leq s_1(\omega) \lambda$-a.e. belongs to $\mathcal{S}_{m,v}$ with $v \in [F_m(t-), F_m(t)]$.

This completes the proof. $\qquad\square$

## 2 Experiments

This section contains information about the conducted experiments. However, the code and details of how to reproduce the results are found in the GitHub repository:

`https://github.com/marcus-nordstrom/optimal-solutions-to-accuracy-and-dice`

### 2.1 Experiment G

#### 2.1.1 Details on experiment

The data consist of 19 patients with 9 ROIs (region of interest) where each ROI has been delineate by 5 separate practitioners. This leads to a total of 855 segmentations. Let $1 \leq r \leq 9$ be an index of the ROIs, and $1 \leq p \leq 19$ be an index of the patients. We then think of each ROI for each patient as a random segmentation $L_{p,r}$ taking values in $\mathcal{S}$. For each such random segmentation, we have access to 5 observations $1 \leq i \leq 5$, and denote each observation with $l_{p,r}^{(i)}$. The marginal function for patient $p$ and ROI $r$ is denoted by $m_{p,r}$ and formed by taking the point-wise average, that is $m_{p,r}(\omega) = \frac{1}{5} \sum_{i=1}^{5} l_{p,r}^{(i)}(\omega), \omega \in \Omega$. In Figure 1, two marginal functions formed this way are illustrated. The experiment we run computes the following list of points:

$$\{ (\|s^{A_{m_{p,r}}}\|_1 / \|m_{p,r}\|_1, \|s^{D_{m_{p,r}}}\|_1 / \|m_{p,r}\|_1) \}_{1 \leq p \leq 19, 1 \leq r \leq 9}. \tag{138}$$

In Figure 5 theses points are illustrated as scatter plots and in Table 1 aggregated statistics of these points are listed.

#### 2.1.2 Comments on license, identifiability and consent

From what we can gather, there is no licence specified by the creators. However, it is specified in the article and in the download link (see below) that the data is free to use for non-commercial purposes. As always is the case with medical image data, identifiable of the patients may be an issue. Different experts and legislators have various opinions as to what lengths researchers has to go in order for the data to be considered anonymized. However, in this case this is not an issue since all of the patients signed informed consent to be part of the data.

#### 2.1.3 Setup instructions

**1. Path:**  Before executing any code, make sure to set the file path to `Exierments_G`. This folder will contain the code together with the data.

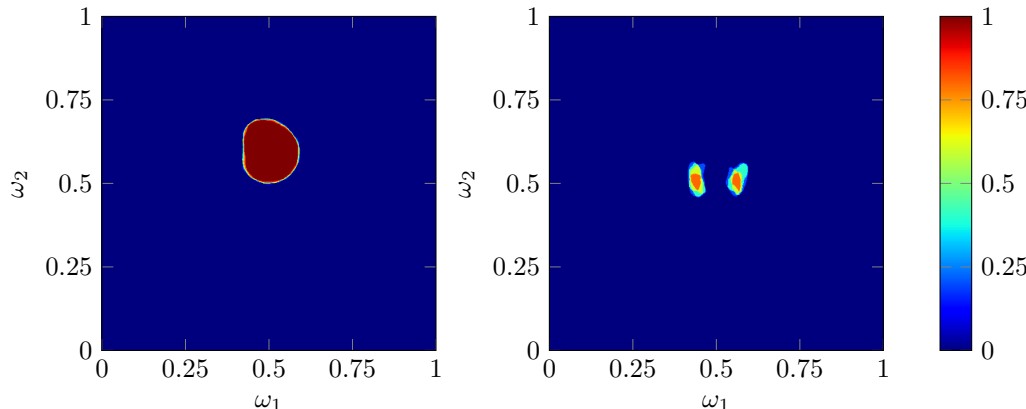

Figure 1: Example of a slice from two marginal functions. The number on the axis are indices associated with the pixels. To the left, an Urinary bladder and to the right Neurovascular bundles. Note that there is almost no disagreement in the first case where as there is lots of disagreement in the second case.

**2. Data:** The data used (Version 1, May 24 2017) can be acquired from the following link.

$$\texttt{https://doi.org/10.5281/zenodo.583096}$$

Scroll down to the box labeled *Files*, click on button labeled *Request access...* and follow the instructions.

Access will be granted provided your fulfill the criteria of requesting the data for academic or educational purposes. When granted, an email will be sent to you with a link. Follow this link and scroll down to the box labeled *Files* again. A list of files, each with a button labeled *"Download"* should now be visible. Download all of the files to the `Experiments_G/dicom` folder and rename `3_03_P(1).zip` to `3_03_P.zip`. When done `Experiments_G/dicom`, should be populated with the following 19 files: `1_01_P.zip, 1_02_P.zip, 1_03_P.zip, 1_04_P.zip, 1_05_P.zip, 1_06_P.zip, 1_07_P.zip, 1_08_P.zip, 2_03_P.zip, 2_04_P.zip, 2_05_P.zip, 2_06_P.zip, 2_09_P.zip, 2_10_P.zip, 2_11_P.zip, 3_01_P.zip, 3_02_P.zip, 3_03_P.zip, 3_04_P.zip`.

**3. Plastimatch:** For the computations, it is necessary to convert the segmentations from the *rtstruct* format to the binary mask format *nrrd*. In this work, Plastimatch version 1.9.3 for Windows 64 which has a BSD-style license is used. Both the installer and license can be found at the following address.

$$\texttt{http://plastimatch.org/}$$

For Ubuntu users, the Plastimatch software is available in the apt-repository. There should be no reason as to why running it on this platform should be a problem, but it has not been tested.

**4. Python:** In this work, Python 3.10.4 is used. It can be downloaded from the following address.

$$\texttt{https://www.python.org/downloads/}$$

Once in the right python environment, the necessary packages can be installed by using the provided requirements file.

```
pip install -r requirements.txt
```

**5. Running the code:** In the `main.py` file, edit the variable `plastimatch_match` so that it is compatible with the install path of Plastimatch. The code is then simply executed with the following.

```
python main.py
```

It will take approximately 30 min on a descent desktop computer and should be no problems running on a laptop. No GPU computations are done. The code will start by unzipping all of the downloaded files to a temporary folder that will be deleted after the run. It will then run Plastimatch to extract a mask for every available segmentation and put the results in `Experiments_G/masks`. Once this is done, the discrete versions of the marginal functions are computed and used to compute the relative volumes. When complete, the results of the experiment can be found under `Experiments_G/results`.

## 2.2 Experiment L

### 2.2.1 Details on experiment

The data consist of 1018 patients with lung nodules which have been delineated by 4 separate practitioners. This leads to a total of 4072 segmentations. Let $1 \leq p \leq 1018$ be an index of the patients. We then think of the ROI for each patient as a random segmentation $L_p$ taking values in $\mathcal{S}$. For each such random segmentation, we have access to 4 observations $1 \leq i \leq 4$, and denote each observation with $l_p^{(i)}$. The marginal function for patient $p$ is denoted by $m_p$ and formed by taking the point-wise average, that is $m_p(\omega) = \frac{1}{4}\sum_{i=1}^4 l_p^{(i)}(\omega), \omega \in \Omega$. In Figure 2, a marginal functions formed this way is illustrated. The experiment we run computes the following list of points:

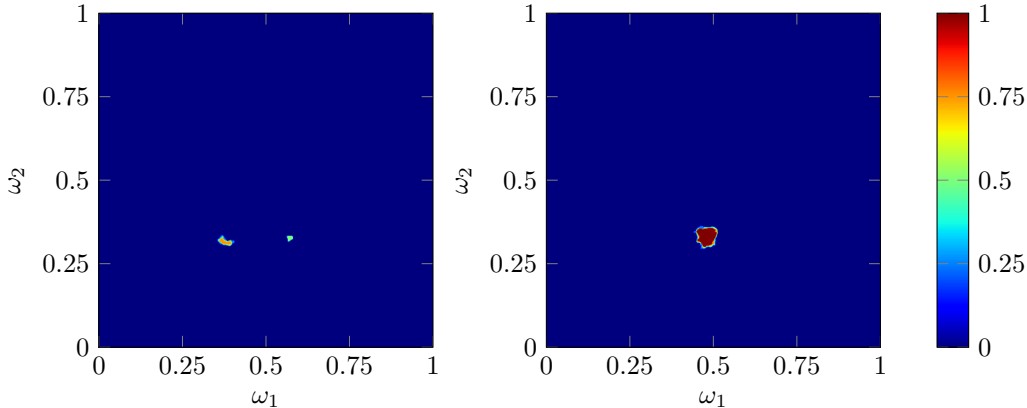

Figure 2: Example of a slice from a marginal function where two lung nodules are visible.

$$\{(\|s^{\mathrm{A}m_p}\|_1/\|m_p\|_1, \|s^{\mathrm{D}m_p}\|_1/\|m_p\|_1)\}_{1\leq p\leq 1018}. \tag{139}$$

In Figure 6 these points are illustrated with scatter plots and in Table 1 aggregated statistics of these points are listed.

### 2.2.2 Comments on license, identifiability and consent

The data is licensed under Creative Commons Attribution 3.0 Unported License.

$$\texttt{https://creativecommons.org/licenses/by/3.0/.}$$

Measures have been taken to anonymize the patient data.

### 2.2.3 Setup instructions

**1. Path:** Before executing any code, make sure to set the file path to `Experiments_L`. This folder will contain the code together with the data when downloaded and processed.

**2. Data:** To get the data set, follow this link

$$\texttt{https://doi.org/10.7937/K9/TCIA.2015.LO9QL9SX}$$

and download the *Radiologist Annotations/Segmentations (XML)* file. In our experiments, version 3 is used, which at the time of writing is the current version. Extract the zip file to the `Experiments_L/xml` folder. Note that the DICOM files are not necessary in order to run the experiments.

**3. Python:** In this work, Python 3.10.4 is used. It can be downloaded from the following address.

$$\texttt{https://www.python.org/downloads/}$$

Once in the right python environment, the necessary packages can be installed by using the provided requirements file.

```
pip install -r requirements.txt
```

**4. Running the code:** When the previous steps are completed, the code is executed by simply running the following.

```
python main.py
```

It will take approximately 30min on a descent desktop computer and should be no problems running on a laptop. No GPU computations are done. The code will use the pilidc library to query annotation data and generate masks. The masks are used to compute the discrete versions of the marginal functions and used to compute the relative volumes. When complete, the results of the experiment can be found under `Experiments_L/results`.