# OpenReview forum: "On Image Segmentation With Noisy Labels: Characterization and Volume Properties of the Optimal Solutions to Accuracy and Dice"
_NeurIPS.cc/2022/Conference — NeurIPS 2022 Accept_

### Official Review · Reviewer_esZf · 2022-07-11

**Rating:** 7
**Confidence:** 1
**Soundness:** 4 excellent
**Presentation:** 3 good
**Contribution:** 4 excellent

**Summary:**

This paper tackled the noisy labels of image segmentation in a theoretical manner. The study is performed on accuracy and dice metrics in medical image segmentation.

**Questions:**

No questions.

**Ethics Review Area:**

["I don’t know"]

**Limitations:**

Yes

**Strengths And Weaknesses:**

From a couple of years ago, dataset curation has been addressed in a wide range. However, most of the findings have not been shared with the community due to the private corporation policies. Thanks to the inconsistency of the  labels detected in the regular datasets used in the wide range of computer vision tasks, efficient solutions have been discovered and proposed. Therefore, this is one of the papers that can address not only segmentation but also edge detection, to name an example. Hence, the paper presents the first discoveries and solutions for the medical image segmentation; It also includes experiments on data from the Gold Atlas project.

Minor errors:
* Page 3 line 123 typo.

---

> ### Author Response · Authors · 2022-08-02
> **We thank esZf for only having positive things to say about the article and communicate that we have corrected a typo.**
>
> Dear esZf,
>
> thanks for taking the time to read and provide comments on our work.
>
> We are very happy that you find our work to provide theoretical insights on noisy image segmentation and agree that our results likely can be extended to edge detection and other areas of computer vision.
>
> Also, thanks for pointing out the typo, we have corrected it.
>
> Best,
> authors.

---

> > ### Comment · Reviewer_esZf · 2022-08-08
> > **I appreciate the authors clarifications,  corrections and additional experiments**
> >
> > I appreciate the authors clarifications, typo correction and additional experiments. I have worked mostly in applied tasks but now I can see more supports to you work.
> >
> > Best wishes,

---

### Official Review · Reviewer_RPBQ · 2022-07-11

**Rating:** 5
**Confidence:** 3
**Soundness:** 2 fair
**Presentation:** 2 fair
**Contribution:** 2 fair

**Summary:**

The authors provided a theoretical analysis of how the optimal segmentation to the metrics Accuracy and Dice can deviate regarding the segmentation volume. In addition, the upper and lower bounds on the volume of optimal segmentation are provided. Unfortunately, although the authors attempt to show the relevance of the theoretical observation in a real dataset, the experiments are not well-introduced and contain many flaws (listed in the weakness below), which makes the experimental results less convincing.

**Questions:**

See the weakness part.

**Limitations:**

Limitations are discussed except for the aspects listed in the weakness.

**Strengths And Weaknesses:**

+ A theoretical analysis of how the volume of optimal segmentation with respect to the performance of Accuracy and Dice could deviate is presented.

Weakness:
- The experiments are not well-introduced, and the results are not well-illustrated. For example, how exactly the marginal function m is computed by "taking the pixel-wise average of the different segmentations" is not clearly introduced? In addition, it is unknown what's the optimal segmentations in the provided dataset (with labels from multiple annotators).
- The selected dataset is small, with only 18 patients, and the ROIs in each case are annotated with unknown medical practitioners.
- It remains unclear how the level of noise in the annotation will affect the evaluation capability of metrics. A conclusion of " the noise has a significant effect on the volume of optimal segmentations" is not practically valuable for detailed applications.
- The authors also mentioned that the shape of the target is related to the effect of noise. Still, only a vague clue is given that spherical structures will be less affected.
- FCN and U-Net-based segmentation methods have been around and popular since 2015, not "during the last years," as stated at the beginning of the Introduction section.

---

> ### Author Response · Authors · 2022-08-02
> **We thank RPBQ for thinking of our theoretical main contribution as a strength and address the negative comments related to the experimental part.**
>
> Dear RPBQ,
>
> thanks for taking the time to read and provide comments on our work.
>
> We are very happy you think of our theoretical contribution as a strength and would like to highlight that this is our main contribution.
> The experimental part is intended as illustration of these results and not a complete experimental investigation.
> We agree that such an experimental investigation would be of interest, but this is beyond the scope of our work.
>
> **The experiments are not well-introduced, and the results are not well-illustrated. For example, how exactly the marginal function m is computed by "taking the pixel-wise average of the different segmentations" is not clearly introduced? In addition, it is unknown what's the optimal segmentations in the provided dataset (with labels from multiple annotators)**.
>
> Details on the experiment (formal expression of pixel-wise average for instance) can be found in the Supplemental Material (section 2.1).
> We realize that this was not communicated in the Experiments section and have therefore changed the sentence
> *Details on the implementation and how to obtain the data is available in the Supplementary Material*,
> found in the end of the section to
> *Details on the experiment, implementation and how to obtain the data is available in the Supplementary Material*.
>
> Regarding that it is unknown what an optimal segmentation is in the provided data set.
> One of the main points in our paper is that what segmentation is optimal depends on what metric is chosen and what kind of noise is present.
> Since we in practice never have access to the exact probabiltiy distribution associated with a random segmentation, we will never know what the optimal segmentations are with respect to the *true marginal function*.
> However, if we use the approximate marginal function obtained by a finite sample approximation, we can compute the optimal segmentations with respect to the estimated marginal function.
> This is what we do in our experiments, and details on how to reproduce these optimal segmentations including python code is found in the Supplementary Material.
>
>
> **The selected dataset is small, with only 18 patients, and the ROIs in each case are annotated with unknown medical practitioners.**
>
> To the best of our knowledge this is the largest and most complete publicly available data set for 3D segmentations with multiple annotators.
> Furthermore, the data set is sufficiently large to illustrate our theoretical results.
> For a complete experimental investigation, however, we agree that a larger data set would be desirable.
>
> Regarding that the annotators are unknown, please see [22] for a complete description of the data set including the annotators.
>
> **It remains unclear how the level of noise in the annotation will affect the evaluation capability of metrics. A conclusion of " the noise has a significant effect on the volume of optimal segmentations" is not practically valuable for detailed applications.**
>
> Under our general assumptions on $m$, the sharp bounds of the volume of the optimal segmentations are wide.
> For a more precise conclusion, stronger assumptions on the structure of $m$ are required.
>
> **The authors also mentioned that the shape of the target is related to the effect of noise. Still, only a vague clue is given that spherical structures will be less affected.**
>
> We mention this in two places, firstly in the end of the Experiments section
> as an observation on our results and secondly in the end of the Conclusion section.
> We have removed the second occurrence since we agree that this is to speculative to be a conclusion.
> Thank you for pointing this out.
>
> **FCN and U-Net-based segmentation methods have been around and popular since 2015, not "during the last years," as stated at the beginning of the Introduction section.**
>
> We agree that 2015 is stretching *during the last years* and have changed it to *since 2015*.
> Thank you for pointing this out.
>
> Best,
> authors.

---

> > ### Comment · Reviewer_RPBQ · 2022-08-04
> > **Not convinced by the presented experiments**
> >
> > I thank the authors for the responses. I am still not convinced about the size of the experimental dataset. There is another dataset LIDC, in which the lung nodule segmentation task contains multiple annotations. Additionally, it is exactly about the number of the data experimented here, or it will be clearer to conclude how the shape of the target would be related to the effect of noise.

---

> > > ### Author Response · Authors · 2022-08-07
> > > **Experiments on LIDC has been added to our work.**
> > >
> > > We agree that the mentioned data set (LIDC) is interesting for our purposes and have therefor edited our work to include experiments with respect to it in addition to the previous experiments. A revised version should now be availible with results in the paper and added code and reproducability details in Supplementary Material. The results are in line with our previous experiments.
> > >
> > > We also updated the first experiments so that it included a 19'th patient which we did not have access to initially. This made no significant difference.
> > >
> > > Best, authors.

---

> > > > ### Comment · Reviewer_RPBQ · 2022-08-08
> > > > **It is great to have the LIDC results**
> > > >
> > > > I appreciate the authors' efforts in the additional results. I am more like an application person, so I rather see it experimentally proofed than theoretically proofed. I might underestimate the value of the presented theoretical analysis.

---

> > > > > ### Author Response · Authors · 2022-08-08
> > > > > **We thank RPBQ for acknowledging our added experiments and for being transparent about bias.**
> > > > >
> > > > > Thank you very much for acknowledging the additional results and for being transparent about your bias.
> > > > >
> > > > > Best, authors.

---

### Official Review · Reviewer_wPye · 2022-07-11

**Rating:** 6
**Confidence:** 3
**Soundness:** 3 good
**Presentation:** 3 good
**Contribution:** 3 good

**Summary:**

It is a theoretical paper that studies two popular performance metrics in medical image segmentation, i.e., Accuracy and Dice while considering the target labels are noisy/random. The noisy labels scenario would be common in segmentation, for example, when different annotators give different boundaries to differ two classes. The authors theoretically prove several theorems to provide a form of optimal solutions to Accuracy and Dice metrics, and derive multiple properties of the optimal solutions, including 1) bounds on the volume of optimal segmentation, 2) relation between the volume of optimal solutions to two metrics. Further, the authors use an experiment to show that the optimal segmentations with respect to Accuracy and Dice deviate from the expected volume of the target.

**Questions:**

Suggestions:
1. As mentioned in Strengths And Weaknesses, the reviewer suggests adding more "informal" descriptions to the notations defined and giving some concrete examples when introducing the theory.
2.  As mentioned in Strengths And Weaknesses, the reviewer suggests adding more information about the significance of the work.

Questions:
1. The theory developed in the paper seems to focus more on the performance evaluation step when using these metrics. Could the current theory help with improving the training steps in some ways?
2. How do we make use of these properties to do better image segmentation? Maybe some examples could be provided.




**Limitations:**

1. I agree with the limitations mentioned by the authors, i.e., the amount of experiments included is limited in the paper, because to demonstrate the usefulness and generality of the theory, results from different medical image segmentation applications may be needed to draw a conclusion.

**Strengths And Weaknesses:**

Originality:
* Strengths: The theoretical analysis on Accuracy and Dice looks original to the reviewer.

Quality:
* Strengths: The main contributions of the paper, i.e., theorems are well-stated and written in a good format, and lemmas and theorems are rigorously proved.

Clarity:
* Strengths: The main ideas of the paper are clearly presented.
* Weaknesses: As the measure theory is widely used in the paper for strictness and generality, sometimes the readers would not be able to get the true meaning/insight of some notations corresponding to their use in the medical segmentation. It is highly recommended that examples (even in an informally speaking way) are given to describe the notations when they first appear in the paper. For example, add some description that $\Omega$ is informally the continuous space for voxels from the data when $n=3$, and give some concrete examples for noisy segmentation $L$. Also, adding a non-measure-theory-style description for Accuracy and Dice would also help understand the paper.

Significance:
* Weaknesses: Authors may need to highlight how their work could help with current medical image segmentation work, and how their insights from the theorems can help the people/scholars in this area, for example, selecting metrics etc.

---

> ### Author Response · Authors · 2022-08-02
> **We thank wPye for thinking of our theoretical work as original with high quality and address the comments related to (1) the measure theoretic style as being complicated to understand and (2) adding information regaring the significance of the work.**
>
> Dear wPye,
>
> thanks for taking the time to read and provide comments on our work.
>
> We are very happy you find our theoretical work to be original, have high quality, and to be clearly presented.
>
> **Suggestion 1:**
>
> Thanks for this suggestion, we agree that these changes might help clarify notations for readers without experience in measure theory, and have expanded the Preliminaries section to address this.
> In particular, we have provided an example of the space of segmentations commonly encountered in practice, commented on the measure theoretic style of the definitions of our metrics and made a reference to Figure 1 as an example of observations of noisy segmentation.
>
> **Suggestion 2:**
>
> Thanks for this suggestion.
> One important consequence of our work comes from the fact that training methods in general have been designed with a certain metric in mind.
> For instance, arguably the two most popular methods in medical image segmentation today include training a neural network with cross-entropy and then thresholding with $1/2$ or training a neural network with soft-Dice (a continuous version of Dice) and thresholding with $1/2$.
> These methods target the metrics Accuracy and Dice, respectively.
> In the literature, it is common to approach problems obtained when using these methods under label noise  by considering more advanced neural network models.
> Our work says that some of these problems, like the volume of predictions not being close to the expected volume, is not a consequence of model choice but rather the choice of metric.
> Hence, instead of experimenting with more advanced models, we sometimes need to change what metric our training method targets in order to address the problem.
>
> We have expanded paragraph two and three in the introduction and changed the conclusion to clarify this connection with training.
>
> **Question 1:**
>
> One way our theory potentially could help with the training step is that it shows that some marginal estimate together with a threshold (in general not $1/2$) theoretically maximizes the Dice score.
> Since cross-entropy is known to be an unbiased estimator of the marginals, a way to maximize Dice is to train a model with cross-entropy and then for each prediction to compute an optimal threshold.
> Finding the optimal threshold can be done efficiently and we provide code for doing this (see the function get\_opt\_D\_seg in main.py in the Supplemental Material).
> This is an attractive alternative to methods that target Dice directly like training a model using soft-Dice and thresholding with 1/2, which has sometimes been reported to be less stable to use during training [21].
> We plan to do an experimental investigation of such a method in future work.
>
>
>
> **Question 2:**
>
> One potential way our work can be used to do better image segmentation is that it provides the theoretical basis for a method that yields optimal segmentations with the volume equal to the expected volume of the target.
> By Theorem 4, when constraining the volume to be equal to the expected volume of the target, that is $v=|m|$, we have that the optimal segmentations with respect to both Accuracy and Dice, are given by the family $\mathcal{S}_{m,|m|}$.
> In practice, such segmentations can be computed by estimating the marginals with a neural network trained with cross-entropy, estimating $|m|$ by computing the sum of all of the voxels used to represent $m$, and then sorting the voxels used to represent $m$ from highest to lowest and assigning $1$ to the first $|m|$ elements and $0$ to the rest.
> We plan to do an experimental investigation of such a method in future work.
>
>
> Best,
> authors.

---

> > ### Comment · Reviewer_wPye · 2022-08-09
> > **Feedback to authors' reply**
> >
> > Thanks very much for addressing my comments per readability and suggestions. I will change my score to 6, considering the current paper after the change. On the one hand, I think theoretical work should be encouraged. On the other hand, I really want to see how the theories can instruct the model design in practice and what differences can be made so that we have better training/testing results.

---

> > > ### Author Response · Authors · 2022-08-09
> > > **We thank wPye for the update and make a comment on the paper style.**
> > >
> > > Thank your for the update. We are very happy you are positive to the response and to the updates.
> > >
> > > We agree that papers with the style of deriving and evaluating new methods can be important and interesting, but due to the wide use of methods targeting Accuracy and Dice under noisy label conditions, we think it is important to improve the theoretical understanding of these methods as well.
> > >
> > > Best, authors.

---

### Official Review · Reviewer_5JSU · 2022-07-11

**Rating:** 6
**Confidence:** 1
**Soundness:** 3 good
**Presentation:** 3 good
**Contribution:** 3 good

**Summary:**

In this paper, the authors present an analysis of the often used performance metrics for image segmentation (Accuracy, Dice) for noisy target labels. They performed a theoretical analysis of the problem and evaluation on a real-world data set. In their results, they provide a characterization of upper and lower bounds on the volume of optimal segmentation.

**Questions:**

None

**Limitations:**

Yes, the authors have addressed the limitations of their work.

**Strengths And Weaknesses:**

The paper is very well written and very detailed. The paper provides a detailed investigation of how noise labels influence the volume of optimal segmentation. This provides interesting insides on how the quality of the manual segmentations influences the final segmentation performance in relation to the state-of-the-art evaluation metrics Dice and accuracy.

---

> ### Author Response · Authors · 2022-08-02
> **We thank 5JSU for only having positive things to say about the article.**
>
> Dear 5JSU,
>
> thanks for taking the time to read and provide comments on our work.
>
> We are very happy that you find our work to be well written, detailed and provide interesting theoretical insights on noisy image segmentation.
>
> Best,
> authors.

---

### Meta-Review · Area_Chair_JnEj · 2022-08-24

**Recommendation:** Accept
**Confidence:** Certain

**Metareview:**

Three knowledgeable referees support accept and a fourth one does not oppose to accept so I recommend Accept.

**Award:**

No

---

### Decision · Program_Chairs · 2022-09-14

Accept